

# On spectrally flowed local vertex operators in AdS$_3$

**Sergio Iguri**[1,2,3⋆] **and Nicolas Kovensky**[4†]

**1** CONICET-Universidad de Buenos Aires, Instituto de Astronomía y Física del Espacio (IAFE),
C. C. 67, Suc. 28, 1428 Buenos Aires, Argentina.
**2** Universidad de Buenos Aires, Facultad de Ciencias Exactas y Naturales,
Ciudad Universitaria, 1428 Buenos Aires, Argentina.
**3** Universidad Abierta Interamericana, Facultad de Arquitectura.
1428 Buenos Aires, Argentina
**4** Institut de Physique Théorique, Université Paris Saclay, CEA, CNRS,
Orme des Merisiers, 91191 Gif-sur-Yvette CEDEX, France.

⋆ siguri@iafe.uba.ar , † nicolas.kovensky@ipht.fr

## Abstract

We provide a novel *local* definition for spectrally flowed vertex operators in the SL(2,ℝ)-WZW model, generalising the proposal of [1] for the singly-flowed case to all $\omega > 1$. This allows us to establish the precise connection between the computation of correlators using the so-called spectral flow operator [1], and the methods introduced recently in [2] based on local Ward identities. We show that the auxiliary variable $y$ used by the authors of [2] arises naturally from a point-splitting procedure in the space-time coordinate. The recursion relations satisfied by spectrally flowed correlators, which take the form of partial differential equations in $y$-space, then correspond to null-state conditions for generalised spectral flowed operators. We highlight the role of certain SL(2,ℝ) discrete module isomorphisms in this context, and prove the validity of the conjecture put forward in [2] for $y$-space structure constants of three-point functions with arbitrary spectral flow charges.



# 1   Introduction

Strings propagating in asymptotically AdS$_3$ geometries and orbifolds thereof constitute one of the fundamental ingredients in our current understanding of some important open questions in quantum gravity in general, and in the microscopic description of black holes in particular. Indeed, they provide a framework in which we can apply powerful computational techniques, such as worldsheet string theory and more generally two-dimensinal conformal field theory (CFT), together with the AdS$_3$/CFT$_2$ duality, and low-dimensional supergravity, to address a wide range of interesting phenomena far beyond the regimes of applicability of supergravity and perturbation theory. The computation of correlation functions in this context has provided crucial insights into the fundamental nature of the holographic duality [3–5], but also non-AdS holography and single-trace $T\bar{T}$ and $J\bar{T}$ deformations of 2d CFTs [6–11], black hole phenomenology [12–20] and even condensed matter physics [21].

When the background configuration is that of global AdS$_3$ with pure NS-NS flux, one can describe the string dynamics in terms of a solvable worldsheet model. The prototypical example is that of type IIB superstring theory on AdS$_3 \times S^3 \times T^4$ (or $K3$). The main ingredient of the corresponding 2d CFT on the worldsheet is the WZW model based on the universal cover of SL(2,$\mathbb{R}$), first studied in [22–25]. Although it is believed to be exactly solvable, many difficulties in computing observables arise from the fact that the target space is both Lorentzian and non-compact. This was understood in [26, 27], where the authors showed that a consistent spectrum and partition function are only obtained upon including the so-called spectrally flowed representations. As opposed to the compact case, the spectral flow automorphisms give rise to new representations, inequivalent to the canonical ones, which encompass in particular a continuum of long string states, *i.e.* those that can reach the asymptotic boundary while remaining finite in energy.

The spectral flow operation is more naturally thought of in the so-called $m$-basis, i.e. where the Cartan current is diagonalized, and the analysis is nicely complemented by considering the parafermionic decomposition [28]. Increasing the spectral flow charge $\omega$ of a given state can be described in terms of what is known as the *spectral flow operator,* whose parafemionic part reduces to a multiple of the identity. The first instances of three-point functions involving spectrally flowed states where computed in [1, 28], using the fact that this spectral flow operator possesses a null descendant.

On the other hand, for holographic purposes, it is necessary to work with operators that are local in spacetime[1], namely, those defined in terms of the $x$-basis. The continuous label $x$, the quantum variable conjugated to $m$, is identified with the (holomorphic) coordinate of the holographic CFT$_2$ living at the AdS$_3$ boundary. Roughly speaking, these local operators are constructed by combining an infinite number of $m$-basis operators, which correspond to

---

[1]Throughout this paper, we follow [1] and refer to $x$-basis vertex operators as *local*. However, we note that computing their spacetime OPEs in full generality remains an interesting open problem.

their spacetime modes. The presence of states with non-trivial spectral flow charge, which are not affine primaries, and their complicated OPEs with the conserved currents render the computation of correlation functions in this model quite complicated.

By using and further developing the techniques of [1], a number of papers have managed to compute certain subfamilies of spectrally flowed local correlation functions [29–35]. However, when attempting to apply these methods to more general correlators, one faces a serious issue since there is no $x$-basis definition of spectrally flowed operators generalizing the $\omega = 1$ case described in [1]. The importance of this statement is made clearer by noting that resorting to our $m$-basis intuition is simply not enough. Indeed, as it was also discussed in [36, 37], and despite our terminology, the relation with the $x$-basis is much more subtle than, say, in the SU(2) case, and, as a consequence, the fusion rules are actually *not* the same for both types of operators.

More recently, an interesting alternative approach was developed in [2,3,38]. The authors made use of a set of *local* Ward identities arising in the spectrally flowed sectors to derive recursion relations satisfied by correlation functions in terms of the spacetime weights $h$ of the different insertions. By introducing the associated conjugate coordinate $y$ as an auxiliary variable, these recursion relations were then recast as partial differential equations, for which a general solution was put forward. The structure of this solution is based on the intuition stemming from the tensionless string scenario, where the theory is believed to be exactly dual to a symmetric product orbifold [36, 37]. In this model, for which the holographic duality is perhaps close to be formally proven [3, 4], correlation functions vanish unless there exists a holomorphic covering map from the string worldsheet to the AdS$_3$ boundary. Although this is not the case in more general situations, the existence of these covering maps underlies the proposal of [2,38].

The solution for the spectrally flowed correlators of the SL(2,$\mathbb{R}$)-WZW model is, however, still to be proven. The main obstacle is that there is no known closed form for the recursion relations alluded above (respectively, partial differential equations in terms of the $y$ variables). So far, they have been derived case by case. Although solving these constraints produces explicit integral expressions for the corresponding correlators, it does so only up to an overall structure constant, which depends on the spectral flow charges and the unflowed SL(2,$\mathbb{R}$) spins of the different insertions, but not on their spacetime weights. Furthermore, since there is no clear relation between the $y$-basis analysis and the spectral flow operator used in [1], it is not well understood why these methods produce results consistent with each other, at least in the limited subset of cases where one can use both.

In this work we fill an important gap in the literature by providing an explicit definition of $x$-basis vertex operators with arbitrary spectral flow charge, namely, for $\omega > 1$, thus generalising the analysis of [1]. This definition is based on a similar point-splitting procedure involving the unflowed vertex and a modified spectral flow operator. It is also recursive in the sense that this spectral flow operator is a flowed version of the original one, albeit by only $\omega - 1$ units. We prove the validity of our proposal by explicitly computing all OPEs with the conserved currents. We also show how to make practical use of this definition for the computation of two- and three-point functions by invoking a modified version of the null-state conditions.

We then establish the precise relation between this formalism and that of [2,38] by showing how the $y$ variable arises from our definition of boundary-local operators in spectrally flowed sectors of the theory. We describe how the partial differential equations satisfied by the correlators in terms of the $y$ variable can be re-interpreted as the null-state conditions associated to the spectral flow operators involved in our construction. Finally, and after discussing the identification series in this context, we show how to fix the explicit form of the structure constants, such that our result coincides with what was conjectured in [2]. Further progress along these lines will be presented in a future paper [39].

## 2 Definitions and conventions

We will mostly follow the conventions of [1,38], and work in the bosonic SL(2,$\mathbb{R}$)-WZW model at level $k > 3$ [40,41]. We also focus directly on the $x$-basis, except for the initial definitions, and omit most anti-holomorphic variables unless necessary.

The holomorphic conserved currents of the model will be denoted $J^a(z)$. They satisfy the OPEs

$$J^a(z)J^b(w) \sim \frac{\eta^{ab}k/2}{(z-w)^2} + \frac{f^{ab}{}_c J^c(w)}{z-w}\,, \tag{1}$$

where $\eta^{+-} = -2\eta^{33} = 2$, $f^{+-}{}_3 = -2$ and $f^{3+}{}_+ = -f^{3-}{}_- = 1$. The energy-momentum tensor and the central charge follow from the Sugawara construction, and are given by

$$T(z) = \frac{1}{k-2} : -J^3(z)J^3(z) + \frac{1}{2}\left[J^+(z)J^-(z) + J^-(z)J^+(z)\right]:\,, \tag{2}$$

and

$$c = \frac{3k}{k-2}\,. \tag{3}$$

The relevant representations of the (holomorphic) zero-mode algebra are as follows. On the one hand, one has the principal discrete series of lowest (highest) weight, spanned by

$$\mathcal{D}_j^{\pm} = \{|j\,m\rangle\,,\ m = \pm j, \pm j \pm 1, \pm j \pm 2, \cdots\}\,, \tag{4}$$

respectively, with $J_0^3|j\,m\rangle = m|j\,m\rangle$. These are unitary representations for any positive real $j$, one being the charge conjugate of the other. There are also the principal continuous series, spanned by

$$\mathcal{C}_j^{\alpha} = \{|j\,m\rangle\,,\ 0 \le \alpha < 1\,,\ j = 1/2 + is\,,\ s \in \mathbb{R}\,,\ m = \alpha, \alpha \pm 1, \alpha \pm 2, \cdots\}\,. \tag{5}$$

It was shown in [26] that a consistent spectrum of the model is built out of continuous and lowest weight representations with

$$\frac{1}{2} < j < \frac{k-1}{2}\,, \tag{6}$$

together with their spectrally flowed images, to be introduced below. Eq. (6) follows from $L^2$ normalization conditions, no-ghost theorems and spectral flow considerations.

In the unflowed sector, the action of the currents on the primary states is given by

$$J_0^3|j\,m\rangle = m|j\,m\rangle\,, \tag{7a}$$

$$J_0^{\pm}|j\,m\rangle = \begin{cases} (m \mp (j-1))|j, m \pm 1\rangle & \text{if } m \ne \mp j \\ 0 & \text{if } m = \mp j\,, \end{cases} \tag{7b}$$

$$J_n^a|j\,m\rangle = 0 \qquad \forall\, n > 0\,. \tag{7c}$$

The corresponding primary vertex operators $V_{jm}(z)$ can be obtained from those of the Euclidean counterpart of the model, namely the $H_3^+$-WZW model [25], by means of the following Mellin-like transform:

$$V_{jm}(z) = \int_{\mathbb{C}} d^2x\, x^{j-m-1} \bar{x}^{j-\bar{m}-1} V_j(x,z)\,, \tag{8}$$

after a well-defined analytical continuation in $j$ is assumed. In the so-called $x$-basis, a generic vertex $V_j(x,z)$ has conformal weight

$$\Delta = -\frac{j(j-1)}{k-2}\,, \tag{9}$$

and is acted upon by the currents as

$$J^a(w)V_j(x,z) \sim \frac{D_j^a V_j(x,z)}{(w-z)},$$

(10)

where

$$D_j^+ = \partial_x, \qquad D_j^3 = x\partial_x + j, \qquad D_j^- = x^2\partial_x + 2jx.$$

(11)

The two-point function is given by

$$\langle V_{j_1}(x_1,z_1)V_{j_2}(x_2,z_2)\rangle = \frac{1}{|z_{12}|^{4\Delta_1}}\left[\delta^2(x_1-x_2)\delta(j_1+j_2-1) + \frac{B(j_1)}{|x_{12}|^{4j_1}}\delta(j_1-j_2)\right],$$

(12)

with

$$B(j) = \frac{2j-1}{\pi}\frac{\Gamma[1-b^2(2j-1)]}{\Gamma[1+b^2(2j-1)]}\nu^{1-2j}, \qquad \nu = \frac{\Gamma[1-b^2]}{\Gamma[1+b^2]}, \qquad b^2 = (k-2)^{-1}.$$

(13)

The three-point function takes the form

$$\langle V_{j_1}(x_1,z_1)V_{j_2}(x_2,z_2)V_{j_3}(x_3,z_3)\rangle = C(j_1,j_2,j_3)\left|\frac{x_{12}^{j_3-j_1-j_2}x_{23}^{j_1-j_2-j_3}x_{31}^{j_2-j_3-j_1}}{z_{12}^{\Delta_1+\Delta_2-\Delta_3}z_{23}^{\Delta_2+\Delta_3-\Delta_1}z_{31}^{\Delta_3+\Delta_1-\Delta_2}}\right|^2,$$

(14)

where the structure constant has a complicated formula given in terms of Barnes double Gamma functions [1, 25]. Among its properties, the following two will be relevant for our purposes:

$$C(j_1,j_2,0) = B(j_1)\delta(j_1-j_2),$$

(15)

$$C\left(j_1,j_2,\frac{k}{2}\right) \sim \delta\left(j_1+j_2-\frac{k}{2}\right),$$

(16)

where the last expressions holds only up to $k$-dependent factors.

At first sight, the complex variable $x$ appears simply as an SL(2,$\mathbb{R}$) version of the isospin variables defined for SU(2) in [42]. However, given that the integrated zero modes of the currents realize the space-time Virasoro modes $L_0$ and $L_{\pm 1}$, and by examining the expressions of the associated differential operators (11), one is led to interpret $x$ as the local coordinate on the boundary theory [22]. According to (12), in the bosonic theory a $z$-integrated vertex operator $V_j(x)$ is identified with a local operator of weight $j$ in the boundary. Conversely, the corresponding boundary Virasoro modes are given by the $m$-basis operators. Indeed, for states in the discrete sector, the transform in Eq. (8) can be inverted, giving

$$V_j(x,z) = \sum_{m,\bar{m}} x^{m-j}\bar{x}^{\bar{m}-j}V_{jm}(z) \sim e^{xJ_0^+ + \bar{x}\bar{J}_0^+}V_{jj}(z)e^{-xJ_0^+ - \bar{x}\bar{J}_0^+}.$$

(17)

The vertex $V_j(x,z)$ is thus realised as $V_{jj}(z) = V_j(x=0,z)$, translated from the origin to $x$. Poles in the integrand of (8) coming from the expansion around $x=0$ ($x=\infty$) are associated to states in the $\mathcal{D}_j^+$ ($\mathcal{D}_j^-$) representation [2,3]. For states in the continuous sector, (17) is modified to account for the fact that, although $m-\bar{m}$ is an integer number, $m+\bar{m}$ can take arbitrary real values, which must be integrated over, giving [38],

$$V_j(x,z) = \frac{i}{(2\pi^2)}\sum_{m-\bar{m}}\int_{-\infty}^{\infty}d(m+\bar{m})\,x^{m-j}\bar{x}^{\bar{m}-j}V_{jm}(z).$$

(18)

States defined in this way satisfy the reflection property

$$V_{1-j}(x,z) = B(1-j)\int d^2x'\,|x-x'|^{4j-4}V_j(x',z),$$

(19)

showing that $B(j)$ defines the reflection coefficient. Note that, although unflowed contiuous states are tachyonic and do not survive the GSO projection in the full superstring construction, their spectrally flowed cousins are actually physical, and describe long string configurations.

In this context, the action of the currents can be compactly written in terms of [23]

$$J(x,z) \equiv e^{xJ_0^+}J^-(z)e^{-xJ_0^+} = J^-(z) - 2xJ^3(z) + x^2J^+(z). \tag{20}$$

Then, the corresponding OPE with the vertex operators reads

$$J(x',z')V_j(x,z) \sim \frac{1}{z'-z}\left[(x-x')^2\partial_{x'} + 2j(x-x')\right]V_j(x,z). \tag{21}$$

## 2.1 Spectral flow

Spectral flow automorphisms of the current algebra (1) are defined as

$$J^\pm(z) \to \tilde{J}^\pm(z) = z^{\pm w}J^\pm(z), \qquad J^3(z) \to \tilde{J}^3(z) = J^3(z) - \frac{k\omega}{2}\frac{1}{z}, \tag{22}$$

where the so-called spectral flow charge $\omega$ is an integer number. Since we work with the universal cover of SL(2,$\mathbb{R}$), holomorphic and anti-holomorphic spectral flow charges must coincide [1]. As mentioned above, the action of (22) on the principal series of SL(2,$\mathbb{R}$) defines representations that are, in general, inequivalent to the canonical ones, and must be considered in order to complete the spectrum.

At the level of primary vertex operators, and for $w > 0$, the mapping introduced in (22) defines the so-called flowed primaries, whose OPEs with the currents take the form

$$J^+(z)V_{jm}^\omega(w) = \frac{(m+1-j)V_{j,m+1}^\omega(w)}{(z-w)^{\omega+1}} + \sum_{n=1}^\omega \frac{(j_{n-1}^+ V_{jm}^\omega)(w)}{(z-w)^n} + \dots, \tag{23a}$$

$$J^3(z)V_{jm}^\omega(w) = \frac{\left(m+\frac{k}{2}\omega\right)V_{jm}^\omega(w)}{(z-w)} + \dots, \tag{23b}$$

$$J^-(z)V_{jm}^\omega(w) = (z-w)^{\omega-1}(m-1+j)V_{j,m-1}^\omega(w) + \dots, \tag{23c}$$

where the ellipsis indicate higher order terms. Similar equations hold for $\omega < 0$ with the roles of $J^+$ and $J^-$ inverted. The operators $V_{jm}^\omega(z)$ are not affine primaries. They are, however, Virasoro primaries with weight

$$\Delta = -\frac{j(j-1)}{k-2} - m\omega - \frac{k}{4}\omega^2, \tag{24}$$

since

$$\tilde{T}(z) = T(z) + \frac{\omega}{z}J^3(z) - \frac{k}{2}\frac{\omega^2}{z^2}. \tag{25}$$

Note that for $\omega > 0$ ($\omega < 0$), independently of the original state, these correspond to lowest (highest) weight states of the zero-mode algebra with spin

$$h = m + \frac{k}{2}\omega, \tag{26}$$

($h = -m - k\omega/2$, respectively). The spectrally flowed affine modules alluded above are built by acting freely with the currents on spectrally flowed primary states. In particular, other states in the corresponding global multiplets are *not* spectrally flowed primaries.

Given that spectral flow is most naturally understood in the $m$-basis, one can formally define vertex operators local in $x$ with non-trivial spectral flow charges by extrapolating the definition (17) [38]. Here, the role of the state at the origin $x = 0$, i.e., the lowest-weight state, is played by the spectrally flowed primary state, that is

$$V_{jh}^{\omega}(x = 0, z) \equiv V_{jm}^{\omega}(z) \qquad (\omega > 0). \tag{27}$$

Since states with both positive and negative $\omega$ contribute to the same vertex, the $x$-basis operator is only defined by the absolute value. Conversely, the state at $V_{jh}^{\omega}(\infty, z)$ is associated to $V_{j,-m}^{-\omega}(z)$.

All properties defining spectrally flowed operators in the $x$-basis can be condensed into the following OPE [43]:

$$
\begin{aligned}
J(x', z') V_{jh}^{\omega}(x, z) \ &\sim \ (x - x')^2 \sum_{n=1}^{\omega+1} \frac{\left(J_{n-1}^+ V_{jh}^{\omega}\right)(x, z)}{(z' - z)^n} + \frac{2h(x - x')V_{jh}^{\omega}(x, z)}{z' - z} + \cdots \\
&\sim \ (x - x')^2 \sum_{n=1}^{\omega+1} \frac{\left(J_{n-1}^+ V_{jh}^{\omega}\right)(x, z)}{(z' - z)^n} + \frac{2h(x - x')V_{jh}^{\omega}(x, z)}{z' - z} + \cdots \\
&\sim \ (x - x')^2 \left\{ \frac{(j - m - 1)V_{j,h+1}^{\omega}(x, z)}{(z' - z)^{\omega+1}} + \sum_{n=2}^{\omega} \frac{\left(J_{n-1}^+ V_{jh}^{\omega}\right)(x, z)}{(z' - z)^n} \right\} \\
&\quad + \frac{(x - x')^2 \partial_x + 2h(x - x')}{z' - z} V_{jh}^{\omega}(x, z) + \cdots.
\end{aligned}
\tag{28}
$$

Although there are higher order poles in the OPEs, we still have

$$\left(J_0^a V_{jh}^{\omega}\right)(x, z) = D_h^a V_{jh}^{\omega}(x, z), \tag{29}$$

with the differential operators (11), while

$$\left(J_{\pm\omega}^{\pm} V_{jh}^{\omega}\right)(x, z) = \left[h - \frac{k}{2}\omega \mp (j - 1)\right] V_{j,h\pm1}^{\omega}(x, z), \tag{30}$$

since $h = m + k\omega/2$.

## 3 Spectrally flowed states in the $x$-basis

Even though local operators defined from $m$-basis spectrally flowed primaries as above capture the correct physics, most of the terms appearing in the formal series expansion generalising (17) and (18) are complicated operators which have no simple $m$-basis expressions. Moreover, the interpretation of $x$-basis correlation functions in terms of infinite sums of the $m$-basis ones is rather subtle. It is then useful to have an independent definition for $V_{jh}^{\omega}(x, z)$, directly built in the $x$-basis.

For the first non-trivial case, $\omega = 1$, such a formula is already known. It was introduced in [1] in terms of the fusion of $V_j(x)$ with the *spectral flow operator* $V_{\frac{k}{2}}(x, z)$. It takes the explicit form

$$V_{jh}^{\omega=1}(x, z) = \lim_{\varepsilon, \bar{\varepsilon} \to 0} \varepsilon^m \bar{\varepsilon}^{\bar{m}} \int d^2 y \, y^{j-m-1} \bar{y}^{j-\bar{m}-1} V_j(x + y, z + \varepsilon) V_{\frac{k}{2}}(x, z), \tag{31}$$

building on the parafermionic decomposition

$$J^3(z) = -\sqrt{\frac{k}{2}} \partial \phi(z), \qquad \phi(z)\phi(w) \sim -\log(z-w), \qquad (32)$$

and

$$V_{jm}^{\omega}(z) \sim \Psi_{jm}(z) e^{\left(m + \frac{k}{2}\omega\right)\sqrt{\frac{2}{k}}\phi(z)}. \qquad (33)$$

Here $\Psi_{jm}(z)$ refers to the parafermionic vertex. Spectral flow only affects the exponential part of the state. It is easy to see that the parafermionic factor in $V_{\frac{k}{2}\frac{k}{2}}$ is a multiple of the identity, i.e. $V_{\frac{k}{2}\frac{k}{2}}$ is a pure exponential in $\phi$ which can be used to construct the spectrally flowed operators. Indeed, for unit spectral flow charge we have

$$\lim_{z' \to z} (z - z')^m V_{jm}(z') V_{\frac{k}{2}\frac{k}{2}}(z) = V_{jm}^{\omega=1}(z). \qquad (34)$$

This explains the presence of the $\varepsilon$-limits in (31), while the $y$ integrals simply select the desired $J_0^3$ eigenstate $V_{jm}(z)$ from the zero-mode representation characterised by $V_j(x+y,z)$ sitting at the origin $x = 0$, according to (8). Note that, in the discrete case, the $y$ integrals can be taken to be holomorphic and anti-holomorphic contour integrals, as opposed to integrals over the whole complex plane.

One of the main results of this paper is to extend (31) to arbitrary spectral flow sectors. The proposed definition reads as follows:

$$V_{jh}^{\omega}(x,z) = \lim_{\varepsilon,\bar{\varepsilon} \to 0} \varepsilon^{m\omega} \bar{\varepsilon}^{\bar{m}\omega} \int d^2 y \, y^{j-m-1} \bar{y}^{j-\bar{m}-1} V_j(x+y, z+\varepsilon) V_{\frac{k}{2}\frac{k}{2}\omega}^{\omega-1}(x,z), \qquad (35)$$

where we recall that $h = m + k\omega/2$. Although we will focus on the proposal (35), there is actually an alternative way to write this. Indeed, as in [1] for the $\omega = 1$ case, one has

$$V_{jh}^{\omega}(x,z) = \lim_{y,\bar{y} \to 0} y^{j-m} \bar{y}^{j-\bar{m}} \int d^2\varepsilon \, \varepsilon^{m\omega-1} \bar{\varepsilon}^{\bar{m}\omega-1} V_j(x+y, z+\varepsilon) V_{\frac{k}{2}\frac{k}{2}\omega}^{\omega-1}(x,z). \qquad (36)$$

One of these expressions is manifestly local in $x$, while the other is manifestly local in $z$. Note that there is an intriguing symmetry between $x$ and $z$ (or rather $z^{\omega}$) which, to some extent, also appears below in Eqs. (50)-(53)[2].

Before proving Eq. (35) in full generality, we first show that it passes a series of non-trivial checks. We will refer to vertex operators of the form $V_{\frac{k}{2}\frac{k}{2}\omega}^{\omega-1}(x,z)$ as generalized spectral flow operators.

First, we note that, for $\omega = 1$, Eq. (35) reduces to (31). Indeed, $V_{\frac{k}{2}\frac{k}{2}}^0(x,z) = V_{\frac{k}{2}}(x,z)$. Second, we evaluate (35) at $x = 0$, and inductively show that $V_{jh}^{\omega}(0,z) = V_{jm}^{\omega}(z)$. We have

$$\begin{aligned}
V_{jh}^{\omega}(0,z) &= \lim_{\varepsilon,\bar{\varepsilon} \to 0} \varepsilon^{m\omega} \bar{\varepsilon}^{\bar{m}\omega} \int d^2 y \, y^{j-m-1} \bar{y}^{j-\bar{m}-1} V_j(y, z+\varepsilon) V_{\frac{k}{2},\frac{k}{2}\omega}^{\omega-1}(0,z) \\
&= \lim_{\varepsilon,\bar{\varepsilon} \to 0} \varepsilon^{m\omega} \bar{\varepsilon}^{\bar{m}\omega} V_{jm}(z+\varepsilon) V_{\frac{k}{2}\frac{k}{2}}^{\omega-1}(z) = V_{jm}^{\omega}(z),
\end{aligned} \qquad (37)$$

where the $y$ integral has become the usual transform to the $m$-basis (8). The inductive step corresponds to setting $V_{\frac{k}{2}\frac{k}{2}\omega}^{\omega-1}(0,z) = V_{\frac{k}{2}\frac{k}{2}}^{\omega-1}(z)$. The final equality corresponds to the extension of (34) to arbitrary $\omega$. Of course, one can relate $V_{jh}^{\omega}(\infty,z)$ to $V_{j,-m}^{-\omega}(z)$ in a similar way.

---

[2]It would be interesting to further explore these features. For this, we thank the referee for pointing out the KZ-BPZ relation discussed in Ref. [42].

A third simple check comes from studying the action of the zero-mode currents on (35). For $J_0^+$, using Eqs. (11) and (29) it first reduces to $\partial_y$ when acting on the spectrally unflowed vertex $V_j(x+y,z+\varepsilon)$ in the integrand, while on the spectral flow operator it gives $\partial_x - \partial_y$, as expected. The action of $J_0^3(x) \equiv J_0^3 - xJ_0^+$, defined in analogy to (20), is slightly more interesting to derive. Using $J_0^3(x) = J_0^3(x+y) + yJ_0^+$, we get

$$
\begin{aligned}
\left(J_0^3 V_{jh}^\omega\right)(x,z) &= \lim_{\varepsilon,\bar\varepsilon\to0} \varepsilon^{m\omega}\bar\varepsilon^{\bar m\omega} \int d^2y\, y^{j-m-1}\bar y^{j-\bar m-1} \times \\
&\qquad \left(j + y\partial_y + \frac{k}{2}\omega\right) V_j(x+y,z+\varepsilon) V_{\frac{k}{2},\frac{k}{2}\omega}^{\omega-1}(x,z) \\
&= \left(m + \frac{k}{2}\omega\right) V_{jh}^\omega(x,z) = h V_{jh}^\omega(x,z),
\end{aligned}
\tag{38}
$$

consistently with (29). A similar computation can be performed for $J_0^-(x)$.

Finally, it is also instructive to see how $J_\omega^+$ achieves the shift $h \to h+1$ in this language. Consider, for simplicity, the operator $V_{jh}^\omega$ evaluated at $z = 0$. Then the action of $J_\omega^+$ on the spectral flow operator inside (35) vanishes because it acts on a state with spectral flow charge $\omega - 1$ as $J_1^+$ on an unflowed state. On the other hand, one needs to be careful when acting on $V_j(x+y,\varepsilon)$ since the worldsheet insertion point is shifted from the origin. By using

$$
J_n^+(0) = \sum_{i=0}^n \binom{n}{i} \varepsilon^{n-i} J_i^+(\varepsilon), \qquad (n > 0),
\tag{39}
$$

which holds for sufficiently small values of $w$, and follows from $J_n^+(\varepsilon) = \oint_w dz (z-w)^n J^+(z)$ [44], the only non-trivial action is from the term proportional to $\varepsilon^\omega$. More precisely, we get

$$
\left(J_\omega^+ V_{jh}^\omega\right)(x,0) = \lim_{\varepsilon,\bar\varepsilon\to0} \varepsilon^{m\omega}\bar\varepsilon^{\bar m\omega} \int d^2y\, y^{j-m-1}\bar y^{j-\bar m-1}\left(\varepsilon^\omega \partial_y\right) V_j(x+y,\varepsilon) V_{\frac{k}{2}\frac{k}{2}\omega}^{\omega-1}(x,0),
\tag{40}
$$

which reproduces (30) upon integrating by parts. The action of $J_{-\omega}^-$ can be obtained analogously by noting that $\left(J_{-\omega}^- V_{\frac{k}{2}\frac{k}{2}\omega}^{\omega-1}\right)(x,z)$ is a null vector.

Let us now prove that spectrally flowed operators as defined in Eq. (35) satisfy the OPE (28) with the currents. We proceed by induction in $\omega$. For the singly-flowed case this was proven in [1]. Assuming the validity of (28) for all operators with spectral flow charges up to $\omega - 1$, we can write

$$
J(x',z') V_{\frac{k}{2}\frac{k}{2}\omega}^{\omega-1}(x,z) \sim \sum_{n=1}^\omega \frac{(x-x')^2}{(z'-z)^n}\left(J_{n-1}^+ V_{\frac{k}{2}\frac{k}{2}\omega}^{\omega-1}\right)(x,z) + \frac{k\omega(x-x')}{z'-z} V_{\frac{k}{2}\frac{k}{2}\omega}^{\omega-1}(x,z).
$$

Then, (35) implies

$$
\begin{aligned}
J(x',z') V_{jh}^\omega(x,z) &\sim \lim_{\varepsilon,\bar\varepsilon\to0} \varepsilon^{m\omega}\bar\varepsilon^{\bar m\omega} \int d^2y\, y^{j-m-1}\bar y^{j-\bar m-1} \left\{ \frac{(x+y-x')^2\partial_y + 2j(x+y-x')}{z'-z-\varepsilon} \right. \\
&\quad \left. + \frac{(x-x')^2(\partial_x-\partial_y)+k\omega(x-x')}{z'-z} \right\} V_j(x+y,z+\varepsilon) V_{\frac{k}{2}\frac{k}{2}\omega}^{\omega-1}(x,z) \\
&\quad + \sum_{n=2}^\omega \frac{(x-x')^2 V_j(x+y,z+\varepsilon)\left(J_{n-1}^+ V_{\frac{k}{2}\frac{k}{2}\omega}^{\omega-1}\right)(x,z)}{(z'-z)^n}.
\end{aligned}
\tag{41}
$$

Let us consider the first term and expand $(z'-z-\varepsilon)^{-1} = \sum_{n=0}^{\infty} \varepsilon^n (z'-z)^{n+1}$. All contributions with $n > \omega$ vanish in the $\varepsilon \to 0$ limit. After integrating by parts, the simple pole in $(z-z')$ takes the form

$$
\lim_{\varepsilon,\bar{\varepsilon}\to 0} \frac{\varepsilon^{m\omega}\bar{\varepsilon}^{\bar{m}\omega}}{z-z'} \int d^2 y \, y^{j-m-1} \bar{y}^{j-\bar{m}-1} \left\{ (m+1-j)(2(x-x')+y) + (2j-2)(x+y-x') \right.
$$
$$
\left. + (x-x')^2 \partial_x + k\omega(x-x') \right\} V_j(x+y, z+\varepsilon) V_{\frac{k}{2}\frac{k}{2}\omega}^{\omega-1}(x,z)
$$
$$
= \frac{(x-x')^2 \partial_x + 2\left(m+\frac{k}{2}\omega\right)(x-x')}{z-z'} V_{jh}^{\omega}(x,z). \tag{42}
$$

where the terms linear in $y$ also vanish in the $\varepsilon \to 0$ limit. Proceeding similarly with the $(z-z')^{\omega+1}$ pole, for which only the $y^{-1}$ term survives due to the presence of the extra $\varepsilon^{\omega}$ factor, we re-obtain Eq.(35) but with $m$ replaced by $m+1$, together with an extra $(j-m-1)$ coefficient, as implied by the action of $J_{\omega}^+ = \tilde{J}_0^+$. Upon setting $z = 0$, the rest of the singular terms have as integrands expressions of the form

$$
\sum_{n=2}^{\omega} \frac{(x-x')^2}{z'^n} \left\{ \varepsilon^{n-1} \partial_y V_j(x+y,\varepsilon) V_{\frac{k}{2}\frac{k}{2}\omega}^{\omega-1}(x,0) + V_j(x+y,\varepsilon) \left( J_{n-1}^+ V_{\frac{k}{2}\frac{k}{2}\omega}^{\omega-1} \right)(x,0) \right\} =
$$
$$
\sum_{n=2}^{\omega} \frac{(x-x')^2}{z'^n} \left\{ \varepsilon^{n-1} \left[ J_0^+(\varepsilon), V_j(x+y,\varepsilon) \right] V_{\frac{k}{2}\frac{k}{2}\omega}^{\omega-1}(x,0) + V_j(x+y,\varepsilon) \left( J_{n-1}^+ V_{\frac{k}{2}\frac{k}{2}\omega}^{\omega-1} \right)(x,0) \right\} .
$$

In order to recover (28) we would like to pull $J_{n-1}^+(0)$ to the left of $V_j(x+y,\varepsilon)$. This is slightly subtle. Even though all the current modes involved are positive and $V_j(x+y,\varepsilon)$ is an affine primary, we do pick up some extra contributions due to the fact that the former come from an expansion of $J^+(z)$ around $z = 0$ instead of $z = \varepsilon$, where $V_j(x+y,\varepsilon)$ is inserted. Again, (39) shows that the set of terms with non-zero powers of $\varepsilon$ obtained in this way exactly cancel the higher order terms coming from the $(z'-\varepsilon)^{-1}$ expansion in the first term of (41), thus yielding the desired result.

Hence, Eq.(35) satisfies (28), and provides an alternative definition for $x$-basis spectrally flowed vertex operators with arbitrary spectral flow charge $\omega$.

## 4 Sample correlation functions and series identifications

In this section we show how the definition (35) can be used for the computation of correlation functions involving spectrally flowed operators. We first focus on two-point functions and then consider a set of three-point functions. Along the way, we clarify the role of the so-called series identifications, i.e. the isomorphisms relating spectrally flowed highest- and lowest-weight representations with consecutive spectral flow charges.

Even though the iteration is not straightforward, our formula (35) is recursive in the sense that it allows us to write a vertex operator with spectral flow charge $\omega$ in terms of two insertions, an unflowed one, and another with charge $\omega - 1$ [3]. As in the proof outlined at the end of the previous section, we will proceed by induction several times along the rest of the paper. For this, we make use of all results derived in [1] for operators with $\omega \le 1$.

---

[3]It would be interesting to further explore the possibility of a recursive definition in which each iteration adds a unit of spectral flow to the vertex operator. We have checked that *naïve* extensions of (35) do not satisfy the expected OPEs with the currents, although the reason for this remains unclear.

## 4.1 Two-point functions

Boundary-local two-point functions of spectrally flowed operators with generic charges were originally derived in [1] by first transforming to the $m$-basis, then using the parafermionic decomposition, and finally going back to the $x$-basis. We will generically refer to this type of procedure as an $m$-basis method, as opposed to the $x$-basis techniques employed in this paper (and also recently in [2, 38]).

Starting from (35), the computation of the two-point function $\left\langle V_{jh}^{\omega}(x,z)V_{j'h'}^{\omega}(x',z')\right\rangle$, which necessarily preserves spectral flow, can be reduced to that of a four-point function of the form

$$\left\langle V_j(x+y,z+\varepsilon)V_{\frac{k}{2}\frac{k}{2}\omega}^{\omega-1}(x,z)V_{j'}(x'+y',z'+\varepsilon')V_{\frac{k}{2}\frac{k}{2}\omega}^{\omega-1}(x',z')\right\rangle. \tag{43}$$

This can be computed exactly due to the fact that the operators $V_{\frac{k}{2}\frac{k}{2}\omega}^{\omega-1}(x,z)$ are built upon $V_{\frac{k}{2}\frac{k}{2}}(z)$, which has a null descendant. For $\omega=1$, the corresponding (unflowed) state satisfies the following null-state condition and simplified Knizhnik-Zamolodchikov (KZ) equation:

$$J_{-1}^{-}\left|\frac{k}{2},\frac{k}{2}\right\rangle = \left(L_{-1}+J_{-1}^3\right)\left|\frac{k}{2},\frac{k}{2}\right\rangle = 0. \tag{44}$$

More generally, for the states created by $V_{\frac{k}{2}\frac{k}{2}\omega}^{\omega-1}(0,0)=V_{\frac{k}{2}\frac{k}{2}}^{\omega-1}(0)$, Eqs. (22) and (25) imply

$$J_{-\omega}^{-}\left|\frac{k}{2},\frac{k}{2},\omega-1\right\rangle = \left(L_{-1}+\omega J_{-1}^3\right)\left|\frac{k}{2},\frac{k}{2},\omega-1\right\rangle = 0, \tag{45}$$

where $|j,m,w\rangle$ stands for the state created by $V_{jm}^w(z)$ when inserted at the origin.

We can use these identities to analyse a slightly more general four-point correlator, which will be useful later on. Consider

$$\left\langle V_{j_1}(x_1,z_1)V_{\frac{k}{2}\frac{k}{2}\omega}^{\omega-1}(x_2,z_2)V_{j_3}(x_3,z_3)V_{j_4h_4}^{\omega-1}(x_4,z_4)\right\rangle =$$

$$\tilde{C}_{\omega}(j_1,j_3,j_4)|\mathcal{F}(x,z)|^2\left|\frac{x_{43}^{(j_1+h_2-j_3-h_4)}x_{42}^{-2h_2}x_{41}^{(h_2+j_3-j_1-h_4)}x_{31}^{(h_4-j_1-h_2-j_3)}}{z_{43}^{(\Delta_3+\Delta_4-\Delta_1-\Delta_2)}z_{42}^{2\Delta_2}z_{41}^{(\Delta_1+\Delta_4-\Delta_2-\Delta_3)}z_{31}^{(\Delta_1+\Delta_2+\Delta_3-\Delta_4)}}\right|^2, \tag{46}$$

where

$$z=\frac{z_{21}z_{43}}{z_{31}z_{42}}, \quad x=\frac{x_{21}x_{43}}{x_{31}x_{42}}, \quad \Delta_2=-\frac{k}{4}\omega^2, \quad h_2=\frac{k}{2}\omega, \quad h_4=j_4+\frac{k}{2}(\omega-1). \tag{47}$$

The identities (45) imply that the insertion of the operators

$$O_{\omega}^{\mathrm{NS}} \equiv \oint_{z_2} dz' J^{-}(x_2,z')\frac{(z'-z_4)^{\omega}}{(z'-z_2)^{\omega}}, \tag{48}$$

$$O_{\omega}^{\mathrm{KZ}} \equiv \partial_{z_2}+\frac{\omega}{z_{24}}\oint_{z_2} dz' J^3(x_2,z')\frac{(z'-z_4)}{(z'-z_2)}, \tag{49}$$

into the correlator on the LHS of (46) gives zero. Let us briefly describe the rationale for constructing $O_{\omega}^{\mathrm{NS}}$ and $O_{\omega}^{\mathrm{KZ}}$. The negative powers of $(z'-z_2)$ in $O_{\omega}^{\mathrm{NS}}$ ensure that we pick up the correct mode $J_{-w}^{-}$ when acting on $V_{\frac{k}{2}\frac{k}{2}\omega}^{\omega-1}(x_2,z_2)$. Conversely, and upon inverting the integration contour, the positive powers of $(z'-z_4)$ avoid picking up residues related to other correlators (most of them unknown) coming from all singular terms in the $J^{-}(x_2,z')V_{j_4h_4}^{\omega-1}(x_4,z_4)$ OPE, see Eq.(28). On the other hand, for $O_{\omega}^{\mathrm{KZ}}$ it turns out that only a unit power of $(z'-z_4)$ is

necessary to avoid these poles. A heuristic way to see this is that we are ultimately interested in taking the limit $x_4 \to \infty$, for which, as discussed above, $V_{j_4 h_4}^{\omega-1}(x_4, z_4) \to V_{j_4, -m_4}^{-(\omega-1)}(z_4)$, with $m_4 = h_4 - \frac{k}{2}(\omega-1)$. Since $\omega \geq 1$, the latter operator only has at most a single pole contribution in the OPE with $J^3(x_2, z')$.

As a consequence, the conformal block $\mathcal{F}(x, z)$ satisfies the differential equations

$$\left[\frac{x}{z^\omega} - \frac{x-1}{(z-1)^\omega}\right] x(x-1)\partial_x \mathcal{F} = \left[\kappa\left(\frac{x^2}{z^\omega} - \frac{(x-1)^2}{(z-1)^\omega}\right) + \frac{2j_1 x}{z^\omega} + \frac{2j_3(x-1)}{(z-1)^\omega}\right] \mathcal{F}, \qquad (50)$$

and

$$-\frac{1}{\omega}\partial_z \mathcal{F} = \frac{x(x-1)}{z(z-1)}\partial_x \mathcal{F} + \left[\frac{j_1}{z} + \frac{j_3}{z-1} + \kappa\left(\frac{x}{z} - \frac{x-1}{z-1}\right)\right]\mathcal{F}, \qquad (51)$$

where $\kappa = h_4 - h_2 - j_1 - j_3$. Similarly to the $\omega = 1$ case considered in [1], this can be solved exactly. We find that, up to a multiplicative constant,

$$\mathcal{F}(x, z) = z^{j_1 \omega}(z-1)^{j_3 \omega} x^{2j_3+\kappa}(x-1)^{2j_1+\kappa} P_\omega(x, z)^{h_2-h_4-j_1-j_3}, \qquad (52)$$

where the last factor contains the polynomials

$$P_\omega(x, z) = z^\omega(x-1) - x(z-1)^\omega. \qquad (53)$$

These polynomials correspond exactly to a subset of those appearing in the four-point function analysis of [38]. In their notation, they correspond to the cases $\tilde{P}_{(\omega-1, \omega-1, 1, 1)}(x, z)$. Moreover, for $\omega = 1$ we simply have $P_1(x, z) = z - x$, showcasing the somewhat unexpected singularity at $z = x$ discussed in [1]. More generally, for $\omega > 1$ we find more complicated singularities at the zeros of (53), which have also been discussed recently in [45].

Eqs. (52) and (53) will allow us to get an explicit expression for (46) up to the constant $\tilde{C}_\omega(j_1, j_3, j_4)$. Regarding this constant, for $\omega = 1$, i.e. for the unflowed four-point function, one can determine it by using the OPE of [25] to factorize the correlator in the $z_{12} \to 0$ limit. Since there is a single state propagating in the corresponding channel, the relevant unflowed three point function reduces to (16) (up to a $j$-independent factor) so that one gets[4] [1]

$$\tilde{C}_1(j_1, j_3, j_4) \sim B\left(\frac{k}{2} - j_1\right)^{-1} C\left(\frac{k}{2} - j_1, j_3, j_4\right) \sim B(j_1) C\left(\frac{k}{2} - j_1, j_3, j_4\right). \qquad (54)$$

However, extending this type of arguments to the flowed sectors is non-trivial. In fact, we note that the alternative methods of [2] did not allow the authors to unambiguously fix the $h$-independent structure constants for spectrally flowed three point functions, whose form was conjectured in order to match with a set of previously known results [1, 29]. In the following subsection we will show that, as a consequence of (35), Eq. (54) actually holds for all $\tilde{C}_\omega(j_1, j_3, j_4)$ with $\omega \geq 1$, i.e.

$$\tilde{C}_\omega(j_1, j_3, j_4) \sim B(j_1) C\left(\frac{k}{2} - j_1, j_3, j_4\right). \qquad (55)$$

In order to further motivate this statement and provide a cross-check for our result in Eqs. (52), (53), we can compare it with the conjecture of [38]. For the case of four-point functions with total spectral flow $\omega_1 + \omega_2 + \omega_3 + \omega_4 \in 2\mathbb{Z}$, this proposal takes the form given their Eq. (3.7), which is analogous to the *unflowed* four-point function when written in terms of the so-called generalized differences $X_{ij}$ which depend explicitly on the $y$ variables and on the polynomials $\tilde{P}(x, z)$ alluded above. The particular set of four-point functions under

---

[4]Here we have ignored a $k$-dependent factor, which will be fixed in Eq. (67) below.

consideration, i.e. those in Eq. (46), is particularly interesting since, as discussed above, the corresponding unflowed conformal block is known exactly. It is given by Eq. (52) for $\omega = 1$. Moreover, the generalized cross-ratio takes the form

$$X \equiv \frac{X_{21}X_{43}}{X_{31}X_{42}} = \frac{x(z-1)^{\omega-1}}{P_{\omega-1}(x,z)} . \tag{56}$$

Upon inserting this into the unflowed conformal block, we find that the proposal of [38] is consistent with our formulae due to the identities

$$X - 1 = \frac{z^{\omega-1}(x-1)}{P_{\omega-1}(x,z)} , \qquad X - z = -\frac{P_\omega(x,z)}{P_{\omega-1}(x,z)} . \tag{57}$$

Hence, we have provided a proof for their conjecture for correlators of the form (46).

Let us come back to the computation of the spectrally flowed two-point function. We take the above results and set $(j_1, j_3, j_4) \rightarrow (j_1, j_2, \frac{k}{2})$, $(h_1, h_3, h_4) \rightarrow (h_1, h_2, \frac{k}{2}\omega)$, $(z_1, z_2, z_3, z_4) \rightarrow (z_1 + \varepsilon_1, z_1, z_2 + \varepsilon_2, z_2)$ and $(x_1, x_2, x_3, x_4) \rightarrow (x_1 + y_1, x_1, x_2 + y_2, x_2)$. Then, the cross-ratios become

$$z = \frac{\varepsilon_1 \varepsilon_2}{z_{12}(z_{12} + \varepsilon_{12})} , \qquad x = \frac{y_1 y_2}{x_{12}(x_{12} + y_{12})} . \tag{58}$$

Note that in this case there are two spectral flow operators involved. Using either of them should lead to the same null-state condition, implying $j_1 = j_2 = j$. Thus, Eq.(43) becomes

$$\tilde{C}_\omega\left(j_1, j_2, \frac{k}{2}\right) \left| \frac{z_{12}^{\frac{k}{2}\omega^2}(z_{12} + \varepsilon_{12})^{-2\Delta_1}z^{j_1\omega}(z-1)^{j_1\omega}}{x_{12}^{k\omega}(x_{12} + y_{12})^{2j_1}[z^\omega(x-1) - x(z-1)^\omega]^{2j_1}} \right|^2 . \tag{59}$$

The holomorphic part of the two-point function is then given by

$$\frac{x_{12}^{-k\omega-m_1-m_2}}{z_{12}^{\left(2\Delta_1 - (m_1+m_2)\omega - \frac{k}{2}\omega^2\right)}} \int dy_1 dy_2\, y_1^{j_1-m_1-1} y_2^{j_1-m_2-1} \left[1 + z^{\frac{\omega}{2}} y_{12} + y_1 y_2\right]^{-2j_1} , \tag{60}$$

where we have momentarily omitted the structure constant, dropped some factors of $z$, which is small in the $\varepsilon_{1,2} \rightarrow 0$ limit, and also rescaled $y_i \rightarrow z^{\frac{\omega}{2}} x_{12} y_i$. We can further ignore the term proportional to $z^{\frac{\omega}{2}}$ in the last factor of the integrand, and change variables to $y_1 = \sqrt{u}v$ and $y_2 = \sqrt{u}v^{-1}$. After re-inserting the anti-holomorphic factors, this integral becomes

$$\int dv\, v^{m_2-m_1-1} \int du\, u^{j_1 - \frac{m_1+m_2}{2} - 1}(u-1)^{-2j_1} = \pi\delta^2(m_1 - m_2)\frac{\gamma(j_1 + m_1)}{\gamma(2j_1)\gamma(1 - j_1 + m_1)} , \tag{61}$$

where we have introduced $\gamma(x) = \Gamma(x)/\Gamma(1 - \bar{x})$. Hence, the two point function contains a contribution of the form

$$\tilde{C}_\omega\left(j_1, j_2, \frac{k}{2}\right) \frac{\pi\delta^2(h_1 - h_2)}{|z_{12}|^{4\left(\Delta_1 - h_1\omega + \frac{k}{4}\omega^2\right)}|x_{12}|^{4h_1}} \frac{\gamma\left(j_1 + h_1 - \frac{k}{2}\omega\right)}{\gamma(2j_1)\gamma\left(1 - j_1 + h_1 - \frac{k}{2}\omega\right)} , \tag{62}$$

which gives the correct bulk term [1] by means of (16) provided Eq. (54) is satisfied for $\omega \geq 1$. Actually, for $j_2 = 1 - j_1$ it turns out that there is an additional distributional solution for Eqs.(50)-(51) given by

$$|\mathcal{F}(x,z)|^2 = \left| z^{j_1\omega}(z-1)^{(1-j_1)\omega}x^{1-2j_1}(x-1)^{2j_1-1}\right|^2 \delta^2\left[P_\omega(x,z)\right] . \tag{63}$$

It is straightforward to check that this solution generates the correct extra contribution, leading to [1]

$$\left\langle V_{j_1 h_1}^\omega(x_1, z_1) V_{j_2 h_2}^\omega(x_2, z_2)\right\rangle = \frac{\delta^2(h_1 - h_2)}{\left|z_{12}^{2\Delta_1} x_{12}^{2h_1}\right|^2} \left[\delta(j_1 + j_2 - 1) + \frac{\pi\delta_{(j_1 - j_2)}B(j_1)\gamma(j_1 + m_1)}{\gamma(2j_1)\gamma(1 - j_1 + m_1)}\right] . \tag{64}$$

## 4.2 Series identifications

In this section we describe how the so-called series identifications work in the $x$-basis, and show that they imply (54) for $\omega \geq 1$, as anticipated above.

It is well known that the inequivalence of irreducible representations with different spectral flow charges holds with the exception of the series identifications, i.e. the affine module isomorphisms given by

$$\hat{\mathcal{D}}_j^{\pm,w} \simeq \hat{\mathcal{D}}_{k/2-j}^{\mp,w\pm 1}. \tag{65}$$

In the $m$-basis, this follows from the spectral flow automorphism, together with the following identity between highest/lowest-weight states:

$$V_{j,-j}^{\omega}(z) = \mathcal{N}(j) V_{\frac{k}{2}-j,\frac{k}{2}-j}^{\omega-1}(z) \qquad \omega \geq 1, \tag{66}$$

where

$$\mathcal{N}(j) = \sqrt{\frac{B(j)}{B\left(\frac{k}{2}-j\right)}}. \tag{67}$$

It can be seen from Eq. (13) that $\mathcal{N}(j) \sim B(j)$ up to $k$-dependent factors[5], and that the $j$-independent factor missing on the RHS of Eq. (16) is precisely $B(j_1)/\mathcal{N}(j_1)$. Thinking about $x$-basis operators as translated from the origin suggests that, in the local basis, Eq. (66) should read

$$V_{j,h=-j+\frac{k}{2}\omega}^{\omega}(x,z) = \mathcal{N}(j) V_{\frac{k}{2}-j,h=\frac{k}{2}-j+\frac{k}{2}(\omega-1)}^{\omega-1}(x,z) \qquad \omega \geq 1. \tag{68}$$

We now show that this follows directly from the definition (35). On the RHS of (68) we have

$$V_{\frac{k}{2}-j,h=\frac{k}{2}-j+\frac{k}{2}(\omega-1)}^{\omega-1}(x,z) = \lim_{\varepsilon \to 0} |\varepsilon|^{(k-2j)(\omega-1)} \int d^2 y |y|^{-2} V_{\frac{k}{2}-j}(x+y,z+\varepsilon) V_{\frac{k}{2},\frac{k}{2}(\omega-1)}^{\omega-2}(x,z). \tag{69}$$

On the other hand, the LHS is of the form

$$\begin{aligned} V_{j,h=-j+\frac{k}{2}\omega}^{\omega}(x,z) &= \lim_{\varepsilon \to 0} |\varepsilon|^{-2j\omega} \int d^2 y' |y'|^{4j-2} V_j(x+y',z+\varepsilon) V_{\frac{k}{2},\frac{k}{2}\omega}^{\omega-1}(x,z) \\ &= \lim_{\varepsilon \to 0} |\varepsilon|^{-2j\omega} \int d^2 y' |y'|^{4j-2} \lim_{\varepsilon' \to 0} |\varepsilon'|^{k(\omega-1)} \int d^2 y |y|^{-2} \\ &\quad \times V_j(x+y',z+\varepsilon) V_{\frac{k}{2}}(x+y,z+\varepsilon') V_{\frac{k}{2},\frac{k}{2}(\omega-1)}^{\omega-2}(x,z), \end{aligned} \tag{70}$$

where we have used (35) twice. We now make use of the OPE formula in the unflowed sector, namely [25]

$$V_{j_1}(x_1,z_1) V_{j_2}(x_2,z_2) \sim \int \frac{d j_3 \, d^2 x_3 \, C(j_1,j_2,j_3) |z_1-z_2|^{2(\Delta_3-\Delta_1-\Delta_2)} V_{1-j_3}(x_3,z_1)}{|x_1-x_2|^{2(j_1+j_2-j_3)} |x_2-x_3|^{2(j_2+j_3-j_1)} |x_1-x_3|^{2(j_1+j_3-j_2)}}. \tag{71}$$

For the specific case $j_1 = j$ and $j_2 = \frac{k}{2}$, this gives

$$V_j(x+y',z+\varepsilon) V_{\frac{k}{2}}(x+y,z+\varepsilon') \tag{72}$$

$$\sim \int d j_3 \, d^2 x_3 \frac{C\left(j,\frac{k}{2},j_3\right) |\varepsilon-\varepsilon'|^{2(\Delta_3-\Delta_j-\Delta_{\frac{k}{2}})} V_{1-j_3}(x_3,z+\varepsilon)}{|y'-y|^{2(j+\frac{k}{2}-j_3)} |x+y-x_3|^{2(\frac{k}{2}+j_3-j)} |x+y'-x_3|^{2(j+j_3-\frac{k}{2})}}$$

$$\sim \int d^2 x_3 \frac{|\varepsilon-\varepsilon'|^{2j} V_{1-\frac{k}{2}+j}(x_3,z+\varepsilon)}{|y'-y|^{4j} |x+y'-x_3|^{2(k-2j)}} \sim \frac{|\varepsilon-\varepsilon'|^{2j}}{|y'-y|^{4j}} V_{\frac{k}{2}-j}(x+y',z+\varepsilon),$$

---

[5]These $k$-dependent factors were ignored in a previous version, as was done in [1]. We thank L. Eberhardt and A. Dei for bringing this to our attention.

where we have used (16) and (19). Inserting this into Eq. (70) and rescaling $y \to y y'$ and $\varepsilon' \to \varepsilon \varepsilon'$, we obtain (69). This holds up to an overall normalization constant, which we fix by asking that at $x = 0$ this coincides with (66), thus proving (68).

We pause for moment and note that, using (68), we can write the spectral flow operator involved in the definition of $V_j^\omega(x, z)$ in Eq. (35) in three different but equivalent ways:

$$V_{\frac{k}{2}\frac{k}{2}\omega}^{\omega-1}(x, z) \sim V_{0, \frac{k}{2}\omega}^{\omega}(x, z) \sim V_{\frac{k}{2}\frac{k}{2}\omega}^{\omega+1}(x, z). \tag{73}$$

The fact that we get three alternative expression is specific to the $j = \frac{k}{2}$ since the related $\tilde{j} = \frac{k}{2} - j = 0$ representation is both highest- and lowest-weight. The middle expression in Eq. (73) has a natural interpretation in terms of the boundary theory as a spacetime *twist* operator.

Let us show that the identity (68) allows us to prove that

$$\tilde{C}_\omega(j, 0, j') = \langle V_j(0,0) V_{\frac{k}{2},\frac{k}{2}\omega}^{\omega-1}(1,1) V_{j',j'+\frac{k}{2}(\omega-1)}^{\omega-1}(\infty,\infty) \rangle = \frac{B(j)}{\mathcal{N}(j)} \delta\left(j + j' - \frac{k}{2}\right), \tag{74}$$

thus extending (16) to the spectrally flowed sectors. Note that this in turn also implies the validity of Eq. (54) beyond the singly-flowed sector. Once again, we proceed by induction. Assuming (74), the derivation presented in the previous section provides the two-point function for states with spectral flow $\omega - 1$. Hence, we have

$$\left\langle V_{\frac{k}{2}-j, -j+\frac{k}{2}\omega}^{\omega-1}(x, z) V_{\frac{k}{2}-j', -j'+\frac{k}{2}\omega}^{\omega-1}(x', z') \right\rangle_{\text{bulk}} = \frac{\delta(j-j') B\left(\frac{k}{2}-j\right)}{(z-z')^{\Delta_j+j\omega-\frac{k}{4}\omega^2}(x-x')^{2h}}. \tag{75}$$

On the other hand, we can compute the same two-point function by using (68) to rewrite, say, the vertex evaluated at $(x, z)$ in terms of the corresponding state with spectral flow charge $\omega$, and insert the definition (35). Then, we find

$$(75) = \mathcal{N}(j)^{-1} \lim_{\varepsilon, \bar{\varepsilon} \to 0} |\varepsilon|^{-2j\omega} \int d^2y\, |y|^{4j-2} \left\langle V_j(x+y, z+\varepsilon) V_{\frac{k}{2},\frac{k}{2}\omega}^{\omega-1}(x, z) V_{\frac{k}{2}-j', -j'+\frac{k}{2}\omega}^{\omega-1}(x', z') \right\rangle. \tag{76}$$

By evaluating the three-point function on the RHS, performing the $y$ integral and taking the $\varepsilon \to 0$ limit, we find that the matching with (75) holds precisely provided the corresponding structure constant is given by (74). While we have derived this result for highest/lowest-weight flowed representations, we expect it to hold in general by analytic continuation in the spins $j$ and $j'$. This completes the flowed two-point function computation outlined in the previous subsection.

We will come back to the series identifications of Eq. (68) in the final section of the paper, highlighting its importance when analysing three-point functions with arbitrary spectral flow charges.

## 4.3 Three-point functions

We now show how to use (35) in the context of the computation of spectrally flowed three point functions. The goal is simply to illustrate how it can be used in some particular cases, leaving a more extensive analysis for future work.

We consider the subset of cases where the spectral flow charges are $(\omega_1, \omega_2, \omega_3) = (\omega, 0, \omega-1)$. This is simple because it can be analized from our result (46), as is seen by writing (only) the operator with the highest spectral flow charge in terms of (35).

Indeed, we have

$$\left\langle V_{j_1 h_1}^{\omega}(x_1, z_1) V_{j_2}(x_2, z_2) V_{j_3 h_3}^{\omega-1}(x_3, z_3) \right\rangle = \lim_{\varepsilon, \bar{\varepsilon} \to 0} \varepsilon^{m_1 \omega} \bar{\varepsilon}^{\bar{m}_1 \omega} \int d^2 y \, y^{j_1 - m_1 - 1} \bar{y}^{j_1 - \bar{m}_1 - 1}$$

$$\times \left\langle V_{j_1}(x_1 + y, z_1 + \varepsilon) V_{\frac{k}{2} \frac{k}{2} \omega}^{\omega-1}(x_1, z_1) V_{j_2}(x_2, z_2) V_{j_3 h_3}^{\omega-1}(x_3, z_3) \right\rangle . \tag{77}$$

Upon inserting Eqs. (52) and (53), the $y$ integral can be performed by rescaling $y \to y z^{\omega} \frac{x_{31} - x_{32}}{x_{21} x_{31}}$ and working at small worldsheet cross-ratio, similarly to what was done in the computation of the two-point function above. This leads to

$$\left\langle V_{j_1 h_1}^{\omega}(x_1, z_1) V_{j_2}(x_2, z_2) V_{j_3 h_3}^{\omega-1}(x_3, z_3) \right\rangle = \left| \frac{z_{12}^{\Delta_3 - \Delta_1 - \Delta_2} x_{23}^{\Delta_1 - \Delta_3 - \Delta_2} x_{13}^{\Delta_2 - \Delta_1 - \Delta_3}}{x_{12}^{h_1 + j_2 - h_3} x_{23}^{h_3 + j_2 - h_1} x_{13}^{h_1 + h_3 - j_2}} \right|^2 \tag{78}$$

$$\times \hat{C}_{\omega}(j_1, j_2, j_3, h_3) \frac{\gamma(h_3 - h_1 + j_2)}{\gamma\left(1 + \frac{k}{2}\omega - j_1 - j_2 - h_3\right) \gamma\left(1 + \frac{k}{2}\omega - j_1 - h_1\right)} . \tag{79}$$

Here $\hat{C}_{\omega}(j_1, j_2, j_3, h_3)$ is a constant that can be computed by considering the propagation of an intermediate state with spectral flow charge $\omega - 1$ in the factorization limit. It then follows from Eq. (74) and the series identification (68) that

$$\hat{C}_{\omega}(j_1, j_2, j_3, h_3) = \mathcal{N}(j) \left\langle V_{\frac{k}{2} - j_1, -j_1 + \frac{k}{2}\omega}^{\omega-1}(0, 0) V_{j_2}(1, 1) V_{j_3 h_3}^{\omega-1}(\infty, \infty) \right\rangle$$

$$= \left\langle V_{j_1, -j_1 + \frac{k}{2}\omega}^{\omega}(0, 0) V_{j_2}(1, 1) V_{j_3 h_3}^{\omega-1}(\infty, \infty) \right\rangle . \tag{80}$$

Hence, this simple application of our formula (35) provides a non-trivial recursion relation for structure constants with charges $(\omega, 0, \omega - 1)$ in terms of the weight $h_1$:

$$\frac{\left\langle V_{j_1 h_1}^{\omega}(0, 0) V_{j_2}(1, 1) V_{j_3 h_3}^{\omega-1}(\infty, \infty) \right\rangle}{\left\langle V_{j_1, -j_1 + \frac{k}{2}\omega}^{\omega}(0, 0) V_{j_2}(1, 1) V_{j_3 h_3}^{\omega-1}(\infty, \infty) \right\rangle} = \frac{\gamma(h_3 - h_1 + j_2)}{\gamma\left(1 + \frac{k}{2}\omega - j_1 - j_2 - h_3\right) \gamma\left(1 + \frac{k}{2}\omega - j_1 - h_1\right)} . \tag{81}$$

The correlators originally derived in [29] indeed satisfy this relation.

Let us finish this section by briefly discussing a slightly more general case, given by the correlators of the form

$$\left\langle V_{j_1 h_1}^{\omega_1}(x_1, z_1) V_{j_2 h_2}^{\omega_2}(x_2, z_2) V_{j_3 h_3}^{\omega_1 - \omega_2 - 1}(x_3, z_3) \right\rangle , \quad \omega_1 > \omega_2 \geq 1 . \tag{82}$$

We note that, due to the fact that all spectral flow charges are non-zero, this type of correlation function is generically not accessible from $m$-basis methods [29]. It can be seen that also in this case (35) leads to further recursion relations, which turn out to be consistent with the results of [38]. The computation is, however, more involved than in the case considered above. Upon writing the leftmost vertex in terms of (35) and using the NS and KZ conditions (45), one obtains a set of unknown contributions coming from the additional poles in the OPEs of the currents with $V_{j_2 h_2}^{\omega_2}$. Fortunately, it turns out that we actually have additional equations as well. Instead of the single operator of $O_{\omega_1}^{\text{NS}}$ defined in (48), the four-point function of interest, namely

$$\left\langle V_{j_1}(x_1, z_1) V_{\frac{k}{2} \frac{k}{2}}^{\omega-1}(x_2, z_2) V_{j_3 h_3}^{\omega_2}(x_3, z_3) V_{j_4 h_4}^{\omega_1 - \omega_2 - 1}(x_4, z_4) \right\rangle \tag{83}$$

is now annihilated by

$$O_{\omega_1, n}^{\text{NS}} \equiv \oint_{z_2} dz' J^-(x_2, z') \frac{(z' - z_4)^n}{(z' - z_2)^{\omega_1 - 1}} , \tag{84}$$

for all $n = \omega_1 - \omega_2 - 1, \ldots \omega_1 - 1$. In summary, together with the KZ condition we have a system of $\omega_2 + 2$ differential equations for the correlator we want to compute where $\omega_2$ unknowns of the form $\left\langle V_{j_1} V_{\frac{k}{2}\frac{k}{2}}^{\omega-1} (J_n^+ V_{j_2 h_2}^{\omega_2}) V_{j_3 h_3}^{\omega_1 - \omega_2 - 1} \right\rangle$ (for $n = 1, \ldots, \omega_2$) appear linearly. These can then be eliminated, leaving once again two differential equations, which allow us to determine this four-point function exactly. Once again, it can be compactly expressed in terms of a more general family of polynomials considered in [38].

## 5 The $y$ variable and point-splitting

There have been several instances where we have seen hints of a connection between our formalism and that of [2, 38]. We now make the connection explicit, thus clarifying how to relate these works with the computations of [1, 28].

So far, several integrals appearing in the computation of different correlation functions suggest a close relation between the integration variable $y$ appearing in our proposal (35) and that introduced in [2, 38] with the so-called $y$-transform. The latter is associated with an object $V_j^\omega(x, y, z)$ defined as a linear combination of the spectrally flowed primary states with all allowed values of $h$ for fixed $j$ and $\omega$, thus mimicking the role of the $x$-basis for unflowed states. To make the relation precise, consider Eq. (35) under the rescaling $y \to y \varepsilon^\omega$, which leads to

$$V_{jh}^\omega(x, z) = \int d^2 y \, y^{j-m-1} \bar{y}^{j-\bar{m}-1} \lim_{\varepsilon, \bar{\varepsilon} \to 0} |\varepsilon|^{2j\omega} V_j(x + y\varepsilon^\omega, z + \varepsilon) V_{\frac{k}{2}, \frac{k}{2}\omega}^{\omega-1}(x, z). \tag{85}$$

Importantly, here we have used the fact that, since the powers of $\varepsilon$ and $\bar{\varepsilon}$ are now independent of $m$ and $\bar{m}$, respectively, we can safely exchange the order of the limiting and integration procedures. We then see that Eq. (85) reproduces exactly the *inverse $y$-transform* as introduced in [2], namely

$$V_{jh}^\omega(x, z) = \int d^2 y \, y^{j-m-1} \bar{y}^{j-\bar{m}-1} V_j^\omega(x, y, z), \tag{86}$$

provided we identify

$$V_j^\omega(x, y, z) \equiv \lim_{\varepsilon, \bar{\varepsilon} \to 0} |\varepsilon|^{2j\omega} V_j(x + y\varepsilon^\omega, z + \varepsilon) V_{\frac{k}{2}\frac{k}{2}\omega}^{\omega-1}(x, z). \tag{87}$$

Eqs. (87) is one of the main results of this paper.

As a first consistency check for this identity, we can show that the reflection symmetry of the unflowed sector given by Eq. (19) implies an analogous property for the $y$-basis operators in the flowed sectors considered in [2]. Indeed, we have

$$B(1-j) \int d^2 y' \, |y - y'|^{4j-4} V_j^\omega(x, y', z)$$

$$= B(1-j) \lim_{\varepsilon, \bar{\varepsilon} \to 0} |\varepsilon|^{2(1-j)\omega} \int d^2 y' \, |x + y\varepsilon^\omega - y'|^{4j-4} V_j(y', z + \varepsilon) V_{\frac{k}{2}, \frac{k}{2}\omega}^{\omega-1}(x, z) \tag{88}$$

$$= \lim_{\varepsilon, \bar{\varepsilon} \to 0} |\varepsilon|^{2(1-j)\omega} V_{1-j}(x + y\varepsilon^\omega, z + \varepsilon) V_{\frac{k}{2}, \frac{k}{2}\omega}^{\omega-1}(x, z) = V_{1-j}^\omega(x, y, z),$$

where in the second line we have changed variables $y' \to y' \varepsilon^\omega + x$, and similarly for $\bar{y}'$.

Moreover, we can precisely show the validity of Eq. (87) by studying the action of the different currents on the RHS. Once again, for simplicity we set $z = 0$. Let us then start by

acting with $J_\omega^+(x) = J_\omega^+$. Using that $J_\omega^+ \sim \varepsilon^\omega J_0^+(\varepsilon)$ when acting on $V_j(x,\varepsilon)$ (where "$(\varepsilon)$" refers to the mode expansion around $z = \varepsilon$), and that it annihilates $V_{\frac{k}{2},\frac{k}{2}\omega}^{\omega-1}(x,0)$, we have

$$\left[J_\omega^+, V_j^\omega(x,y,0)\right] = \lim_{\varepsilon,\bar\varepsilon \to 0} |\varepsilon|^{2j\omega}\left[\varepsilon^w \frac{\partial}{\partial(y\varepsilon^\omega)}\right] V_j(x+y\varepsilon^\omega,\varepsilon) V_{\frac{k}{2}\frac{k}{2}\omega}^{\omega-1}(x,0) = \partial_y V_j^\omega(x,y,0), \quad (89)$$

so that $J_\omega^+$ generates translations in $y$. Similarly, we can act with $J_{-\omega}^-(x)$. From the action on the unflowed vertex with shifted insertions, we have $J_{-\omega}^-(x) \sim \varepsilon^{-\omega} J_0^-(x)(\varepsilon)$, thus leading to

$$\left[J_{-\omega}^-(x), V_j^\omega(x,y,0)\right] = \left(2jy + y^2 \partial_y\right) V_j^\omega(x,y,0). \quad (90)$$

Note that $J_{-\omega}^-(x)$ annihilates the spectral flow operator, since it equals $\tilde{J}_{-1}^-(x)$. Unlike in the previous case, the vanishing of $\left[\tilde{J}_{-1}^-(x), V_{\frac{k}{2}\frac{k}{2}\omega}^{\omega-1}(x,0)\right]$ is non-trivial, and it is a consequence of the fact that the latter has a null descendent. Finally, it is simpler to compute the action of the Cartan current, which acts non-trivially on both the unflowed vertex and the spectral flow operator, giving

$$\left[J_0^3(x), V_j^\omega(x,y,0)\right] = \left(j + \frac{k}{2}\omega + y\partial_y\right) V_j^\omega(x,y,0), \quad (91)$$

since $J_0^3(x) = J_0^3(x+y\varepsilon^\omega) + y\varepsilon^\omega J_0^+$. Consequently, we find that these results exactly match the properties described in [2], proving that Eq. (87) provides an alternative definition for the $y$-basis operators, and relating it to the $x$-basis formalism originally used in [1], and further extended in the present work.

The expression introduced in Eq. (87) allows us to highlight the role of holomorphic covering maps in this context. The proposal of [2] for $y$-basis three-point functions can be understood in terms of the properties of certain holomorphic maps from the worldsheet to the AdS$_3$ boundary. When the spectral flow charges involved in a given correlator are such that the associated covering map $\Sigma$ actually exists, it behaves near an insertion point $z$ as

$$\Sigma(z+\varepsilon) = x + a\varepsilon^\omega + \cdots, \quad (92)$$

for some parameter $a$ and small $\varepsilon$. Comparing this with $V_j(x+y\varepsilon^\omega, z+\varepsilon)$ inside Eq.(87), we see that the parameters $a$ are very special points in the $y$-plane: the $y$-basis correlators of [2] present divergences whenever the $y$ variable associated to one of the insertions approaches the coefficients defining a related covering map. We will come back to this in [39].

## 5.1 Recursion relations from null-state conditions

We now show how the definition (87) provides a reformulation of the recursion relations for correlation functions of spectrally flowed operators, which become differential equations in the $y$-basis, derived case by case in [2,3].

We first recall their derivation from the local Ward identities of the SL(2,$\mathbb{R}$)-WZW model. Consider a generic $n$-point correlator $\left\langle V_{j_1 h_1}^{\omega_1}(x_1,z_1) \ldots V_{j_n h_n}^{\omega_n}(x_n,z_n) \right\rangle$. Upon inserting the conserved currents $J^a(z)$ and using the OPEs (28), one obtains cumbersome linear relations with the following three main ingredients:

- the usual differential operators $D_a$ in Eq. (11) acting on the original correlator,

- a total of $\sum_{i=1}^n (\omega_i - 1)$ new unknown correlators of the form

$$\left\langle V_{j_1 h_1}^{\omega_1}(x_1,z_1) \ldots (J_p^+ V_{j_i h_i}^{\omega_i})(x_i,z_i) \ldots V_{j_n h_n}^{\omega_n}(x_n,z_n) \right\rangle, \quad (93)$$

for $i = 1,\ldots,n$ and $p = 1,\ldots,\omega_i - 1$ ($p = 0$ would correspond to the action of $D_a$) and

- correlators where one of the spacetime weights has been shifted by one unit, that is,

$$\left\langle V^{\omega_1}_{j_1 h_1}(x_1, z_1) \ldots V^{\omega_i}_{j_i h_i \pm 1}(x_i, z_i) \ldots V^{\omega_n}_{j_n h_n}(x_n, z_n) \right\rangle, \tag{94}$$

for $i = 1, \ldots, n$. For the upper sign, and up to a coefficient, these actually correspond to the $p = \omega_i$ cases of Eq. (93).

One can actually solve for the unknowns (93). Eq. (28) implies that inserting $J^-(x_i, z)$, with $x_i$ one of the insertion points, must lead to a regular expression as $z \to z_i$. Moreover, the coefficients of the first $\omega_i - 1$ regular terms vanish identically. In other words, one has

$$\begin{aligned}
&\left\langle J^-(x_i, z) V^{\omega_1}_{j_1 h_1}(x_1, z_1) \ldots V^{\omega_n}_{j_n h_n}(x_n, z_n) \right\rangle \\
&= (z - z_i)^{\omega_i - 1} (j_i + m_i - 1) \left\langle V^{\omega_1}_{j_1 h_1}(x_1, z_1) \ldots V^{\omega_i}_{j_i h_i - 1}(x_i, z_i) \ldots V^{\omega_n}_{j_n h_n}(x_n, z_n) \right\rangle + \cdots,
\end{aligned} \tag{95}$$

for $i = 1, \ldots, n$, and with $m_i = h_i - \frac{k}{2} \omega_i$. This gives a total of $\sum_{i=1}^{n} \omega_i$ linear equations, which, after solving for the above unknowns, renders a system of recursion relations – also involving differential operators in $x_i$ – for the actual primary correlators we want to compute.

At the level of three-point functions, the dependence on the insertion points $x_i$ and $z_i$ are fixed by conformal invariance on the worldsheet and on the AdS$_3$ boundary, respectively. However, the resulting recursion relations are still difficult to solve in general. For instance, for the specific case $\omega_1 = \omega_2 = \omega_3 = 1$ they read

$$\begin{aligned}
&(m_1 + j_1 - 1) \left\langle V^1_{j_1, h_1 - 1} V^1_{j_2 h_2} V^1_{j_3 h_3} \right\rangle + (h_2 + h_3 - h_1) \left\langle V^1_{j_1 h_1} V^1_{j_2 h_2} V^1_{j_3 h_3} \right\rangle \\
&= (m_2 - j_2 + 1) \left\langle V^1_{j_1 h_1} V^1_{j_2, h_2 + 1} V^1_{j_3 h_3} \right\rangle + (m_3 - j_3 + 1) \left\langle V^1_{j_1 h_1} V^1_{j_2 h_2} V^1_{j_3 h_3 + 1} \right\rangle,
\end{aligned} \tag{96}$$

and similarly for $1 \leftrightarrow 2$ and $1 \leftrightarrow 3$, and where we have set $x_1 = z_1 = 0$, $x_2 = z_2 = 1$ and $x_3 = z_3 = \infty$. Note that this corresponds to one of the cases that can not be computed in general by using the methods of [1] and [29], since all operators have non-zero spectral flow.

The main motivation for introducing the $y$ variable in [2] was that it transforms these recursion relations into partial differential equations, as indicated by Eqs. (89), (90) and (91). In our example, the relation (96) becomes

$$\left[ y_1(y_1 - 1)\partial_{y_1} + 2y_1 j_1 + (y_2 - 1)\partial_{y_2} + (y_3 - 1)\partial_{y_3} + \kappa \right] \left\langle V^1_{j_1}(y_1) V^1_{j_2}(y_2) V^1_{j_3}(y_3) \right\rangle = 0, \tag{97}$$

with $\kappa = \frac{k}{2} - j_1 + j_2 + j_3$. The correlator on the right-hand side of (97) stands for

$$\left\langle V^1_{j_1}(0, y_1, 0) V^1_{j_2}(1, y_2, 1) V^1_{j_3}(\infty, y_3, \infty) \right\rangle. \tag{98}$$

As discussed in [38], the original correlator can then be obtained as

$$\begin{aligned}
&\left\langle V^{\omega_1}_{j_1}(x_1, y_1, z_1) V^{\omega_2}_{j_2}(x_2, y_2, z_2) V^{\omega_3}_{j_3}(x_3, y_3, z_3) \right\rangle = \frac{x_{21}^{h_3^0 - h_1^0 - h_2^0} x_{31}^{h_2^0 - h_1^0 - h_3^0} x_{32}^{h_1^0 - h_2^0 - h_3^0}}{z_{21}^{\Delta_1^0 + \Delta_2^0 - \Delta_3^0} z_{31}^{\Delta_1^0 + \Delta_3^0 - \Delta_2^0} z_{32}^{\Delta_2^0 + \Delta_3^0 - \Delta_1^0}} \times \\
&\left\langle V^{\omega_1}_{j_1}\left(0, y_1 \frac{x_{32} z_{21}^{\omega_1} z_{31}^{\omega_1}}{x_{21} x_{31} z_{32}^{\omega_1}}, 0\right) V^{\omega_2}_{j_2}\left(1, y_2 \frac{x_{31} z_{21}^{\omega_2} z_{32}^{\omega_2}}{x_{21} x_{32} z_{31}^{\omega_2}}, 1\right) V^{\omega_3}_{j_3}\left(\infty, y_3 \frac{x_{21} z_{31}^{\omega_3} z_{32}^{\omega_3}}{x_{31} x_{32} z_{21}^{\omega_3}}, \infty\right) \right\rangle.
\end{aligned} \tag{99}$$

A comprehensive set of linear relations for three-point functions similar to that of Eq. (97) were derived in [2]. This was done for sufficiently low values of the spectral flow charges $\omega_i$, leading the authors to conjecture a general solution, up to an overall $h$-independent constant,

based on the theory of holomorphic covering maps. However, no general expression for these differential equations is known, and this conjecture remains to be proven.

We now show that the definition in Eq. (87) provides a new, perhaps more direct perspective on the origin of these constraints. We do this in the particular case considered above, that is, when all three insertions have unit spectral flow. In light of Eq. (87), this corresponds to a specific limit of a six-point function, where three of the insertions are spectral flow operators. More precisely, we have

$$
\lim_{\varepsilon_1,\varepsilon_2,\varepsilon_3\to 0}\left\langle\prod_{i=1}^{3}|\varepsilon_i|^{2j_i}V_{j_i}(x_i+y_i\varepsilon_i,z_i+\varepsilon_i)V_{\frac{k}{2}}(x_i,z_i)\right\rangle .
\tag{100}
$$

As reviewed above, the spectral flow operators have a null descendant. Consequently, the six point function on the right-hand side of Eq. (100) must satisfy three differential equations, associated to the corresponding null-state conditions. Using the OPEs of unflowed $x$-basis vertex operators with the affine currents, we see that the condition coming from the $V_{\frac{k}{2}}(x_1,z_1)$ insertion takes the following form:

$$
\left\{\sum_{i=2}^{3}\frac{x_{i1}+y_i\varepsilon_i}{z_{i1}+\varepsilon_i}\left[(x_{i1}+y_i\varepsilon_i)\frac{\partial_{y_i}}{\varepsilon_i}+2j_i\right]+\frac{x_{i1}}{z_{i1}}\left[x_{i1}\left(\partial_{x_i}-\frac{\partial_{y_i}}{\varepsilon_i}+k\right)\right]\right.
$$
$$
\left.+y_1\left(y_1\partial_{y_1}+2j_1\right)\right\}\left\langle\prod_{i=1}^{3}V_{j_i}(x_i+y_i\varepsilon_i,z_i+\varepsilon_i)V_{\frac{k}{2}}(x_i,z_i)\right\rangle=0 .
\tag{101}
$$

Here, the first two terms come from the action of $J^-(x_1)$ on $V_{j_i}(x_i+y_i\varepsilon_i,z_i+\varepsilon_i)$ and $V_{\frac{k}{2}}(x_i,z_i)$, respectively (and with $i=2,3$), while the final term comes from the action on $V_{j_1}(x_1+y_1\varepsilon_1,z_1+\varepsilon_1)$, and where we have already cancelled some $\varepsilon_1$ factors. We are only interested in the $\varepsilon_{1,2,3}\to 0$ limit of this relation. Fortunately, the divergent terms in the first line of Eq. (101), which scale as $\varepsilon_{2,3}^{-1}$, cancel exactly. Hence, (87) leads to the conclusion that the correlator $\left\langle V_{j_1}^1(x_1,y_1,z_1)V_{j_2}^1(x_2,y_2,z_2)V_{j_3}^1(x_3,y_3,z_3)\right\rangle$ is annihilated by the differential operator

$$
\sum_{i=2}^{3}\frac{x_{i1}}{z_{i1}}\left[2y_i\partial_{y_i}+x_{i1}\left(\partial_{x_i}-\frac{\partial_{y_i}}{z_{i1}}\right)+2j_i+k\right]+y_1\left(y_1\partial_{y_1}+2j_1\right) .
\tag{102}
$$

Upon using (99), this becomes exactly the condition written in Eq. (97). An analogous computation holds for $1\leftrightarrow 2$ and $1\leftrightarrow 3$.

In other words, the recursion relations of [2,3] can be understood as arising form the null-state conditions associated to the spectral flow operator appearing in the alternative definition in Eq. (87).

## 5.2 Fixing the structure constants

An explicit proposal for $y$-basis three-point functions with arbitrary spectral flow charges was put forward in [2]. The authors conjectured that, for the odd parity case, namely $\omega_1+\omega_2+\omega_3\in 2\mathbb{Z}+1$,

$$
\left\langle V_{j_1}^{\omega_1}(y_1)V_{j_2}^{\omega_2}(y_2)V_{j_3}^{\omega_3}(y_3)\right\rangle=C_{\boldsymbol\omega}(j_1,j_2,j_3)X_{123}^{\frac{k}{2}-j_1-j_2-j_3}\prod_{i=1}^{3}X_i^{-\frac{k}{2}+j_1+j_2+j_3-2j_i} ,
\tag{103}
$$

while for the even parity case, i.e. when $\omega_1+\omega_2+\omega_3\in 2\mathbb{Z}$,

$$
\left\langle V_{j_1}^{\omega_1}(y_1)V_{j_2}^{\omega_2}(y_2)V_{j_3}^{\omega_3}(y_3)\right\rangle=C_{\boldsymbol\omega}(j_1,j_2,j_3)X_{\emptyset}^{j_1+j_2+j_3-k}\prod_{i<\ell}X_{i\ell}^{j_1+j_2+j_3-2j_i-2j_\ell} ,
\tag{104}
$$

where, for any subset $I \subset \{1, 2, 3\}$, $X_I(y_1, y_2, y_3)$ is defined as

$$X_I(y_1, y_2, y_3) = \sum_{i \in I:\ \varepsilon_i = \pm 1} P_{\boldsymbol{\omega} + \sum_{i \in I} \varepsilon_i e_i} \prod_{i \in I} y_i^{\frac{1 - \varepsilon_i}{2}}. \tag{105}$$

Here we have omitted the right-moving dependence. In eq. (105), $\boldsymbol{\omega} = (\omega_1, \omega_2, \omega_3)$ and the coefficients $P_{\boldsymbol{\omega}}$ are fixed based on certain holomorphic covering maps. Although it was not proven, the dependence on the $y_i$-variables contained in the *generalized differences* $X_I$ was strongly motivated from a case by case study of local Ward identities. On the other hand, the overall constant factors $C_{\boldsymbol{\omega}}(j_1, j_2, j_3)$, were argued to be very simply related to the unflowed structure constants $C(j_1, j_2, j_3)$ based on the comparison with results previously computed in [29] for restricted families of correlators, a relation that was shown to pass several consistency checks. More precisely, the authors proposed that

$$C_{\boldsymbol{\omega}}(j_1, j_2, j_3) = \begin{cases} C(j_1, j_2, j_3), & \text{if} \quad \omega_1 + \omega_2 + \omega_3 \in 2\mathbb{Z}, \\ \mathcal{N}(j_1) C\left(\frac{k}{2} - j_1, j_2, j_3\right), & \text{if} \quad \omega_1 + \omega_2 + \omega_3 \in 2\mathbb{Z} + 1, \end{cases} \tag{106}$$

which is unambiguously defined since

$$\mathcal{N}(j_1) C\left(\frac{k}{2} - j_1, j_2, j_3\right) = \mathcal{N}(j_2) C\left(j_1, \frac{k}{2} - j_2, j_3\right) = \mathcal{N}(j_3) C\left(j_1, j_2, \frac{k}{2} - j_3\right). \tag{107}$$

In this final section we prove that, provided the $y_i$-dependence of Eqs. (103) and (104) is correct, $C_{\boldsymbol{\omega}}(j_1, j_2, j_3)$ is indeed given by (106). We do so by using the series identification $y$-basis formula leading to (68) by means of (86). More precisely, we have

$$V_{j,j+\frac{k}{2}\omega}^{\omega}(x, z) = V_j^{\omega}(x, y = 0, z), \qquad V_{j,-j+\frac{k}{2}\omega}^{\omega}(x, z) = \lim_{y \to \infty} y^{2j} V_j^{\omega}(x, y, z), \tag{108}$$

so that

$$\lim_{y \to \infty} |y|^{4j} V_j^{\omega}(x; y; z) = \mathcal{N}(j) V_{\frac{k}{2} - j}^{\omega - 1}(x; y = 0; z). \tag{109}$$

This leads to the following identity:

$$\lim_{y_3 \to \infty} |y_3|^{4j_3} \left\langle V_{j_1}^{\omega_1}(y_1) V_{j_2}^{\omega_2}(y_2) V_{j_3}^{\omega_3}(y_3) \right\rangle = \mathcal{N}(j_3) \left\langle V_{j_1}^{\omega_1}(y_1) V_{j_2}^{\omega_2}(y_2) V_{\frac{k}{2} - j_3}^{\omega_3 - 1}(y_3 = 0) \right\rangle, \tag{110}$$

which will give us a recursion relation for $C_{\boldsymbol{\omega}}(j_1, j_2, j_3)$. The latter being a constant, we can freely set $y_1 = y_2 = 0$. For the even parity case, both in the LHS and in the RHS of (110), the product of $X_I$ factors reduces to

$$P_{\boldsymbol{\omega}}^{j_1 + j_2 + j_3 - k} P_{\boldsymbol{\omega} + e_1 + e_2}^{j_3 - j_1 - j_2} P_{\boldsymbol{\omega} + e_2 - e_3}^{j_1 - j_2 - j_3} P_{\boldsymbol{\omega} + e_1 - e_3}^{j_2 - j_3 - j_1}, \tag{111}$$

while for the odd parity case they gives

$$P_{\boldsymbol{\omega} + e_1 + e_2 - e_3}^{\frac{k}{2} - j_1 - j_2 - j_3} P_{\boldsymbol{\omega} + e_1}^{-j_1 + j_2 + j_3 - \frac{k}{2}} P_{\boldsymbol{\omega} + e_2}^{j_1 - j_2 + j_3 - \frac{k}{2}} P_{\boldsymbol{\omega} - e_3}^{j_1 + j_2 - j_3 - \frac{k}{2}}. \tag{112}$$

In both cases, we find that Eq. (110) holds iff

$$C_{\boldsymbol{\omega}}(j_1, j_2, j_3) = \mathcal{N}(j_3) C_{\boldsymbol{\omega} - e_3}\left(j_1, j_2, \frac{k}{2} - j_3\right). \tag{113}$$

An analogous computation can be carried out for either of the first two insertions instead. We conclude that the constants $C_{\boldsymbol{\omega}}(j_1, j_2, j_3)$ can be obtained recursively starting from the unflowed case. Indeed, using (107) it is straightforward to see that $C_{\boldsymbol{\omega}}(j_1, j_2, j_3) = C_{\boldsymbol{\omega} - e_i - e_j}(j_1, j_2, j_3)$ for all $i, j = 1, 2, 3$, so that the result only depends on the overall parity of $\omega_1 + \omega_2 + \omega_3$ and gives either $C_0(j_1, j_2, j_3) = C(j_1, j_2, j_3)$ or $C_{e_1}(j_1, j_2, j_3) = \mathcal{N}(j_1) C(\frac{k}{2} - j_1, j_2, j_3)$ [1], as stated in (106).

Although we have proven (106) for discrete representations, we expect that it holds also for the continuous series by analytic continuation in $j$ [1, 46].

# 6 Discussion

Let us summarise the main results of this paper. We have obtained an $x$-basis definition of spectrally flowed local operators in the SL(2,$\mathbb{R}$)-WZW model with an arbitrary spectral flow charge. This is given in Eq. (35), which generalises the point-splitting expression and the concept of spectral flow operator introduced in [1,26] for the singly flowed case. Then, we identified the auxiliary variable $y$ involved in this definition with that introduced recently in [2,38], thus clarifying the relation between the two approaches. In particular, the differential equations satisfied by correlation functions in $y$-space previously derived from local Ward identities were re-interpreted as null state conditions for the generalised spectral flow operators appearing in (35).

Our alternative approach provides a new set of tools for proving the conjecture of [2] for all spectrally flowed three-point functions. As a first step, we have been able to fix the corresponding $y$-independent part of the structure constants, namely Eq. (106). It would be extremely interesting to use (35) to obtain a closed form for the full set of recursion relations satisfied by the local correlators of spectrally flowed vertex operators. This would allow us to elucidate the nature of the so-called generalised differences $X_I$ appearing in Eqs. (103) and (104), and also in spectrally flowed four-point functions [38]. Significant progress in this direction will be addressed in [39].

## Acknowledgements

It is a pleasure to thank Davide Bufalini, Soumangsu Chakraborty, Andrea Dei, Lorenz Eberhardt, Monica Güica, Emil Martinec, Stefano Massai, Sylvain Ribault, Julián Toro and David Turton for interesting discussions and comments.

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
