# Peer review of "On spectrally flowed local vertex operators in AdS$_3$"

_SciPost Physics, doi:SciPost Phys. 13, 115 (2022)_

## Round 1 · Referee Report · Anonymous (Referee 1) · 2022-8-27

Report

The paper studies spectrally flowed vertex operators and their correlation functions in the SL(2,R) Wess-Zumino-Witten (WZW) model. Building on the parafermion decomposition, the authors identify a new definition for spectrally flowed vertex operators carrying explicit dependence on the spacetime coordinate x.

The result is important for various reasons: 1) It extends to generic spectral flow $\omega$ the techniques introduced in Refs [1] and [28] for $\omega=1$. 2) It clarifies the relation of a recent series of works [2,38,43] with the more traditional approach of [1] and [28] to compute correlators of spectrally flowed vertex operators. 3) Assuming the dependence on the spacetime conformal weight is the one proposed in [2], it shows that the $j_i$-dependent overall normalisation of the three-point functions must indeed be the one proposed in [2].

We suggest a number of minor changes that we believe would improve the quality of the paper.

  1. It is not clear to us what the authors exactly mean by the adjective "local". Does it stand for "carrying an explicit dependence on the spacetime position label x"? Locality in spacetime, in the sense of 2D CFT, would require to show that the vertex operators satisfy some kind of spacetime OPE. However, this is still an open problem.

  2. In the second line of page 4, [40] is probably not the correct reference. Ref. [40] mainly deals with $k=3$, which is a boundary value the authors are not focusing on.

  3. At the beginning of page 3 the authors mention that "there is no x-basis definition of spectrally flowed operators beyond the $\omega = 1$ case described in [1]". While we may understand what the authors have in mind, strictly speaking, this is not true: the x-basis vertex operators can be defined in terms of their OPE's with the SL(2,R) currents.

  4. At various places in the text, the authors mention that the proposal of [2] for three-point functions "relies heavily" on the properties of holomorphic covering maps. However, in the proposal for 3-point functions covering maps are not needed. Covering maps are instead used in the proposal of [38] for four-point functions.

  5. What do the authors mean by the tilde in eq. (34)? That the equality is true up to an unknown normalisation factor? Would this normalisation factor depend on j, m, k? Same question for eqs. (54). This is probably clear to the authors, but not obvious for the average reader.

  6. There is maybe a typo in eq. (39). J^+_n in the left-hand-side should be placed at \epsilon, while J^+_i in the right-hand-side should be at 0. It would be useful to the reader if the authors could provide a derivation or a reference for eq. (39).

  7. We do not understand the need of the derivation starting below eq. (40), proving the OPE (28). Given that the authors already showed that eq. (35) has the correct OPEs with the SL(2,R) currents $J^a(z)$ and that $J(x',z')$ is just a linear combination of the $J^a(z')$'s, isn't this derivation redundant?

  8. The notation used in eq. (45) for spectrally flowed states has not been introduced before.

  9. For a reader not familiar with the computation the authors are carrying out, the expression "the correlator (46) is annihilated by" above eq. (48) may be confusing. It would be hepful if the authors could reformulate the sentence or maybe add an equation with a specific correlator being equal to zero.

  10. How is the relative normalisation in eq. (65) fixed? Is this consistent with eq. (63)? Is this expression exact in $k$?

  11. An extra + or - label in eq. (65) and similar equations would be useful, in order to distinguish vertex operators of highest/lowest-weight representations. Since the derivation of eq. (73) relies on eq. (65), is eq. (73) only valid for a specific choice of highest/lowest-weight representations? If not, why?

  12. It is not clear to us how $\tilde C_w(j_1, j_2, j_3)$ is defined. By eqs. (46), (52) and (53) together? Or is it defined as a specific three-point function?

  13. What do $V_{conf}$ and the subscript "bulk" mean in eq. (74) ? More generally, it is not clear to us how eq. (74) follows from (73) with $\omega \to \omega -1$. It would be useful if the authors could add a few intermediate steps in the derivation of eq. (73).

  14. The change of variables adopted in the second line of (86) is probably the inverse of the one mentioned, namely $y' \to \varepsilon^{-w}(y'-x)$.

Requested changes

  1. It is not clear to us what the authors exactly mean by the adjective "local". Does it stand for "carrying an explicit dependence on the spacetime position label x"? Locality in spacetime, in the sense of 2D CFT, would require to show that the vertex operators satisfy some kind of spacetime OPE. However, this is still an open problem.

  2. In the second line of page 4, [40] is probably not the correct reference. Ref. [40] mainly deals with $k=3$, which is a boundary value the authors are not focusing on.

  3. At the beginning of page 3 the authors mention that "there is no x-basis definition of spectrally flowed operators beyond the $\omega = 1$ case described in [1]". While we may understand what the authors have in mind, strictly speaking, this is not true: the x-basis vertex operators can be defined in terms of their OPE's with the SL(2,R) currents.

  4. At various places in the text, the authors mention that the proposal of [2] for three-point functions "relies heavily" on the properties of holomorphic covering maps. However, in the proposal for 3-point functions covering maps are not needed. Covering maps are instead used in the proposal of [38] for four-point functions.

  5. What do the authors mean by the tilde in eq. (34)? That the equality is true up to an unknown normalisation factor? Would this normalisation factor depend on j, m, k? Same question for eqs. (54). This is probably clear to the authors, but not obvious for the average reader.

  6. There is maybe a typo in eq. (39). J^+_n in the left-hand-side should be placed at \epsilon, while J^+_i in the right-hand-side should be at 0. It would be useful to the reader if the authors could provide a derivation or a reference for eq. (39).

  7. We do not understand the need of the derivation starting below eq. (40), proving the OPE (28). Given that the authors already showed that eq. (35) has the correct OPEs with the SL(2,R) currents $J^a(z)$ and that J(x',z') is just a linear combination of the $J^a(z')$'s, isn't this derivation redundant?

  8. The notation used in eq. (45) for spectrally flowed states has not been introduced before.

  9. For a reader not familiar with the computation the authors are carrying out, the expression "the correlator (46) is annihilated by" above eq. (48) may be confusing. It would be hepful if the authors could reformulate the sentence or maybe add an equation with a specific correlator being equal to zero.

  10. How is the relative normalisation in eq. (65) fixed? Is this consistent with eq. (63)? Is this expression exact in $k$?

  11. An extra + or - label in eq. (65) and similar equations would be useful, in order to distinguish vertex operators of highest/lowest-weight representations. Since the derivation of eq. (73) relies on eq. (65), is eq. (73) only valid for a specific choice of highest/lowest-weight representations? If not, why?

  12. It is not clear to us how $\tilde C_w(j_1, j_2, j_3)$ is defined. By eqs. (46), (52) and (53) together? Or is it defined as a specific three-point function?

  13. What do $V_{conf}$ and the subscript "bulk" mean in eq. (74) ? More generally, it is not clear to us how eq. (74) follows from (73) with $\omega \to \omega -1$. It would be useful if the authors could add a few intermediate steps in the derivation of eq. (73).

  14. The change of variables adopted in the second line of (86) is probably the inverse of the one mentioned, namely $y' \to \varepsilon^{-w}(y'-x)$.

  • validity: -
  • significance: -
  • originality: -
  • clarity: -
  • formatting: -
  • grammar: -

Author:  Nicolas Kovensky  on 2022-09-27  [id 2852]

(in reply to Report 1 on 2022-08-27)
Category:
remark
answer to question
correction

We thank the referee for their detailed reading of our manuscript. We address their comments, questions and suggestions below:

1) Page 1, footnote added "Throughout this paper, we follow \cite{Maldacena:2001km} and refer to $x$-basis vertex operators as \textit{local}. However, we note that computing their spacetime OPEs in full generality remains an interesting open problem."

2) Ref.[40] in the second line of page 4 is included in order to point the reader towards the explanation of exactly why the models with k smaller than or equal to 3 are to be treated differently. In that sense, the reference is correct.

3) We agree with the referee that a definition of $x$-basis spectrally flowed operators can be formally given in terms of the OPEs with the currents. There are, however, two issues with this definition. First, it does not fix the normalization. Second, these OPEs contain a large number of terms that, unfortunately, remain unknown. In this paper we have, in a sense, bypassed both issues by providing a point-splitting definition involving auxiliary degenerate fields --- the "generalized" spectral flow operators --- which give an additional handle for the computation of correlation functions. For clarity, we have changed the text from "there is no $x$-basis definition of spectrally flowed operators beyond the $\w=1$ case described in [1]" to "there is no $x$-basis definition of spectrally flowed operators generalizing the $\w=1$ case described in [1]".

4) We agree with the referee about this point, although we do consider that holomorphic covering maps are an important ingredient even for the computation of three-point functions. Nevertheless, we have replaced "relies heavily" by "can be understood in terms of" in the corresponding sentence.

5) We thank the referee for his comment about Eq.(34). Here and in Eq.(33) there was a typo: Eq.(33) comes with a "~" sign because the parafermionic decomposition is written as shown up to an (irrelevant) k-dependent factor, while Eq.(34) can be taken as a formal definition, which fixes the normalization, so that we have now written it with an "=" sign. As for Eq.(54), we have added a footnote at the end of page 12, where we clarify that the overall factor we ignore here is fixed by the discussion around Eq.(66).

6) We thank the referee for his thorough reading. However, there is no typo in Eq.(39). We have added a line of text and a reference below this equation, clarifying, in particular, that it holds only for when $\epsilon$ is sufficiently close to zero (which is all that is needed for the derivation).

7) Between the proposed definition in Eq.(35) and the proof below Eq.(40), we have only computed the action of a restricted set of current modes, namely the zero-modes, J_w^+ and J_{-w}^-. The proof below Eq.(40) is thus needed in order to show that the OPEs with the full currents are as in Eq.(28).

8) We thank the referee for pointing this out. We have added a line below Eq.(45) explaining our notation.

9) We thank the referee for their comment. We have modified the corresponding text, which now reads as follows: "The identities (45) imply that the insertion of the operators (...) into the correlator on the LHS of (46) gives zero."

10) The relative normalization in Eq.(65) is exact. Indeed, it is fixed by consistency with the two-point function in Eq.(63).

11) We thank the referee for their comment. In Eq.(65) there is no need for an extra "+" or "-" superscript since the values of the (unflowed) projections "m" determine the representations that the different states belong to. As for Eq.(73), we have followed the referee's suggestion and added the following comment in the text below: "While we have derived this result for highest/lowest-weight flowed representations, we expected it to hold in general by analytic continuation in the spins $j$ and $j'$."

12) Yes, this structure constant is defined by Eqs.(46), (52) and (53) together.

13) We thank the referee for their comment. We have added a comment above Eq.(74) clarifying the procedure. On the other hand, the subscript "bulk" refers to the discussion below Eq.(61) and singles out the contribution to the two-point function that is non-zero when j1=j2 (as opposed to the one relevant for j1=1-j2).

14) We thank the referee for their question, but, as stated in the text, the new variable is " u = y' \epsilon^w + x ", which we then rename " y' ".

---

## Round 1 · Referee Report · Sylvain Ribault (Referee 2) · 2022-9-16

Report

This technical article lays the basis for proving recent conjectural results by Dei and Eberhardt on spectrally flowed correlation functions in the $SL(2,\mathbb{R})$ WZW model, using methods introduced in the classic works of Maldacena and Ooguri.

While Dei and Eberhardt focus on Ward identities and therefore determine correlation functions up to an overall constant, the present article uses auxiliary correlation functions that involve degenerate fields. This method was originally introduced by Teschner in the context of Liouville theory. However, the degenerate fields that are relevant for computing spectrally flowed correlation functions are rather special, and were only determined by Maldacena and Ooguri in the case $\omega =1$ --- whereas the spectral flow number $\omega$ can take any integer value.

Analytic bootstrap methods based on degenerate fields are appropriate for computing correlation functions in exactly solvable CFTs such as the $SL(2,\mathbb{R})$ WZW model. They are much preferable to Coulomb gas integrals, which only work in particular cases. The present article succeeds in its (relatively modest) aims and derives the three-point structure constant in some special cases.

This suggests that it would be interesting to understand all the model's degenerate fields more systematically: their fusion rules, how they behave under spectral flow. This may well shed light on the puzzling observation that $x$-basis fusion rules differ from $m$-basis fusion rules.

I did not check the calculations in detail. The soundness of the method, and the agreement of the results with previous work, suggest that they are correct. I have a few improvements to suggest:

  1. The expression "series identification" is awkward and non-standard, a clearer formulation would be welcome, even if it was longer.

  2. In Section 3, a couple of calculations involving parafermions are hinted at: "$V_{\frac{k}{2},\frac{k}{2}}$ is a pure exponential", "the trivial extension of (34)". This would be clearer with explicit formulas.

  3. The formula (36) is non-trivial and intriguing. It would be good to move it to the main text (rather than a footnote) and to comment it.

  4. The notion of "frame" is non-standard terminology, and appears only twice. It would be good to eliminate it.

  5. In the formulas (36) and (53), and to some extent in the differential equations (50) and (51), there is an intriguing symmetry between $z$ and $x$. It would be interesting to explain it. Is it related to the KZ-BPZ relation of A. B. Zamolodchikov and V. A. Fateev? [Operator algebra and correlation functions in the two-dimensional wess-zumino su(2) x su(2) chiral model, Sov. J. Nucl. Phys. 43 (1986) 657–664]

  6. The sentence "Eq. (54) actually holds for all $C_\omega(j_1,j_3,j_4)$ with $\omega\geq 1$", and "(54) for $\omega\geq 1$", are unclear: a formula would be better.

  7. The main results (35) and (85) could be better emphasized. (Put in boxes?) Their interpretation in terms of fusion of degenerate fields could be discussed.

  8. It would be interesting to discuss why it does not work to use the original spectral flow operator of Maldacena and Ooguri to add spectral flow unit by unit, and why it is better (or even necessary) to find an operator that adds $\omega$ units of spectral flow.

  • validity: -
  • significance: -
  • originality: -
  • clarity: -
  • formatting: -
  • grammar: -

Author:  Nicolas Kovensky  on 2022-09-27  [id 2853]

(in reply to Report 2 by Sylvain Ribault on 2022-09-16)
Category:
remark
answer to question
suggestion for further work

We thank Prof. S. Ribault for his comments and suggestions, to which we respond below:

1) We have changed "SL(2,R) series identifications" for "certain SL(2,R) discrete module isomorphisms" in the abstract, and also included a clarification at the end of the first paragraph of section 4.

2) We have clarified the sense in which V_{k/2 k/2} is a "pure exponential" in the text above Eq.(34) and removed the word "trivial" while discussing Eq.(37), whose second line is explicitly the extension of Eq.(34) to w>1.

3 and 5) We thank the referee for his comments. We have moved the footnote to the text, and included a small discussion of Eq.(36), including Ref.[41].

4) We thank the referee for his suggestion. We have reformulated the text and removed the term "frame" from the manuscript.

6) We thank the referee for his suggestion. We have added Eq.(55).

7) We have highlighted the importance of Eqs.(35) and (86) as suggested by the referee. Regarding their interpretation in terms of the fusion rules of degenerate fields, this can be understood by looking at Eqs.(34), (37) and (74). We have decided to leave a more extensive discussion on this topic for future work.

8) We have included a footnote in page 10 addressing the referee's comment on this subject.

---

## Round 2 · Referee Report · Anonymous (Referee 1) · 2022-9-27

Report

We thank the authors for their resubmission and for clarifying various points.

We would suggest to include 1803.04423, together with [40] as a reference on page 4. The two papers appeared on the same day on the arXiv and both investigate the tensionless limit k=3.

We have no further suggestions and we recommend the manuscript for publication.

---

## Round 2 · Author Response

Dear editor, we hereby resubmit a second version of the manuscript, which includes all changes suggested by the referees.

---

## Round 2 · List of Changes

Warnings issued while processing user-supplied markup:

  • Inconsistency: plain/Markdown and reStructuredText syntaxes are mixed. Markdown will be used.
    Add "#coerce:reST" or "#coerce:plain" as the first line of your text to force reStructuredText or no markup.
    You may also contact the helpdesk if the formatting is incorrect and you are unable to edit your text.

Here is the list of chages, addressing the comments and suggestions from both referees:

1.1) Page 1, footnote added "Throughout this paper, we follow \cite{Maldacena:2001km} and refer to $x$-basis vertex operators as \textit{local}. However, we note that computing their spacetime OPEs in full generality remains an interesting open problem."

1.3) We agree with the referee that a definition of $x$-basis spectrally flowed operators can be formally given in terms of the OPEs with the currents. There are, however, two issues with this definition. First, it does not fix the normalization. Second, these OPEs contain a large number of terms that, unfortunately, remain unknown. In this paper we have, in a sense, bypassed both issues by providing a point-splitting definition involving auxiliary degenerate fields --- the "generalized" spectral flow operators --- which give an additional handle for the computation of correlation functions. For clarity, we have changed the text from "there is no $x$-basis definition of spectrally flowed operators beyond the $\w=1$ case described in [1]" to "there is no $x$-basis definition of spectrally flowed operators generalizing the $\w=1$ case described in [1]".

1.4) We agree with the referee about this point, although we do consider that holomorphic covering maps are an important ingredient even for the computation of three-point functions. Nevertheless, we have replaced "relies heavily" by "can be understood in terms of" in the corresponding sentence.

1.5) We thank the referee for his comment about Eq.(34). Here and in Eq.(33) there was a typo: Eq.(33) comes with a "~" sign because the parafermionic decomposition is written as shown up to an (irrelevant) k-dependent factor, while Eq.(34) can be taken as a formal definition, which fixes the normalization, so that we have now written it with a "=" sign. As for Eq.(54), we have added a footnote at the end of page 12, where we clarify that the overall factor we ignore here is fixed by the discussion around Eq.(66).

1.6) We thank the referee for his thorough reading. However, there is no typo in Eq.(39). We have added a line of text and a reference below this equation.

1.8) We thank the referee for pointing this out. We have added a line below Eq.(45) explaining our notation.

1.9) We thank the referee for their comment. We have modified the corresponding text, which now reads as follows: "The identities (45) imply that the insertion of the operators (...) into the correlator on the LHS of (46) gives zero."

1.11) We thank the referee for their comment. In Eq.(65) there is no need for an extra "+" or "-" superscript since the values of the (unflowed) projections "m" uniquely determine the representations that the different states belong to. As for Eq.(73), we have followed the referee's suggestion and added the following comment in the text below: "While we have derived this result for highest/lowest-weight flowed representations, it is expected it to hold in general by analytic continuation in the spins $j$ and $j'$."

1.13) We thank the referee for their comment. We have added a comment above Eq.(74) clarifying the procedure. On the other hand, the subscript "bulk" refers to the discussion below Eq.(61) and singles out the contribution to the two-point function that is non-zero when j1=j2 (as opposed to the one relevant for j1=1-j2).

2.1) We have changed "SL(2,R) series identifications" for "certain SL(2,R) discrete module isomorphisms" in the abstract, and also included a clarification at the end of the first paragraph of section 4.

2.2) We have clarified the sense in which V_{k/2 k/2} is a "pure exponential" in the text above Eq.(34) and removed the word "trivial" while discussing Eq.(37), whose second line is explicitly the extension of Eq.(34) to w>1.

2.3 and 2.5) We thank the referee for his comments. We have moved the footnote to the text, and included a small discussion of Eq.(36), including Ref.[41].

2.4) We thank the referee for his suggestion. We have reformulated the text and removed the term "frame" from the manuscript.

2.6) We thank the referee for his suggestion. We have added Eq.(55).

2.7) We have highlighted the importance of Eqs.(35) and (86) as suggested by the referee. Regarding their interpretation in terms of the fusion rules of degenerate fields, this can be understood by looking at Eqs.(34), (37) and (74). We have decided to leave a more extensive discussion on this topic for future work.

2.8) We have included a footnote in page 10 addressing the referee's comment on this subject.

--

We have also updated Refs.[17] and [44], which are now published.

---

## Round 3 · Author Response

Ref.[41] added.

---

## Editorial Decision

published